# PROBING BERT IN HYPERBOLIC SPACES

**Boli Chen**[1,3*], **Yao Fu**[2*]
**Guangwei Xu**[1], **Pengjun Xie**[1], **Chuanqi Tan**[1], **Mosha Chen**[1], **Liping Jing**[3†]
[1]Alibaba Group [2]University of Edinburgh [3]Beijing Jiaotong University
`boli.cbl@alibaba-inc.com, yao.fu@ed.ac.uk,`
`{kunka.xgw, chengchen.xpj}@alibaba-inc.com,`
`{chuanqi.tcq, chenmosha.cms}@alibaba-inc.com,`
`lpjing@bjtu.edu.cn`

## ABSTRACT

Recently, a variety of probing tasks are proposed to discover linguistic properties learned in contextualized word embeddings. Many of these works implicitly assume these embeddings lay in certain metric spaces, typically the Euclidean space. This work considers a family of geometrically special spaces, the hyperbolic spaces, that exhibit better inductive biases for hierarchical structures and may better reveal linguistic hierarchies encoded in contextualized representations. We introduce a *Poincaré probe*, a structural probe projecting these embeddings into a Poincaré subspace with explicitly defined hierarchies. We focus on two probing objectives: (a) dependency trees where the hierarchy is defined as head-dependent structures; (b) lexical sentiments where the hierarchy is defined as the polarity of words (positivity and negativity). We argue that a key desideratum of a probe is its sensitivity to the existence of linguistic structures. We apply our probes on BERT, a typical contextualized embedding model. In a syntactic subspace, our probe better recovers tree structures than Euclidean probes, revealing the possibility that the geometry of BERT syntax may not necessarily be Euclidean. In a sentiment subspace, we reveal two possible meta-embeddings for positive and negative sentiments and show how lexically-controlled contextualization would change the geometric localization of embeddings. We demonstrate the findings with our Poincaré probe via extensive experiments and visualization[1].

## 1 INTRODUCTION

Contextualized word representations with pretrained language models have significantly advanced NLP progress (Peters et al., 2018a; Devlin et al., 2019). Previous works point out that abundant linguistic knowledge implicitly exists in these representations (Belinkov et al., 2017; Peters et al., 2018b;a; Tenney et al., 2019). This paper is primarily inspired by Hewitt & Manning (2019) who propose a *structural probe* to recover dependency trees encoded under squared Euclidean distance in a syntactic subspace. Although being an implicit assumption, there is no strict evidence that the geometry of these syntactic subspaces should be Euclidean, especially under the fact that the Euclidean space has intrinsic difficulties for modeling trees (Linial et al., 1995).

We propose to impose and explore different inductive biases for modeling syntactic subspaces. The hyperbolic space, a special Riemannian space with constant negative curvature, is a good candidate because of its tree-likeness (Nickel & Kiela, 2017; Sarkar, 2011). We adopt a generalized *Poincaré Ball*, a special model of hyperbolic spaces, to construct a *Poincaré probe* for contextualized embeddings. Figure 1 (A, B) give an example of a tree embedded in the Poincaré ball and compare the Euclidean counterparts. Intuitively, the volume of a Poincaré ball grows *exponentially with its radius*, which is similar to the phenomenon that the number of nodes of a full tree grows *exponentially with its depth*. This would give "enough space" to embed the tree. In the meantime, the volume of the Euclidean ball grows polynomially and thus has less capacity to embed tree nodes.

---

[*]Equal contribution. Work was done during an internship at Alibaba DAMO Academy.
[†]Corresponding author.
[1]Our results can be reproduced at `https://github.com/FranxYao/PoincareProbe`.

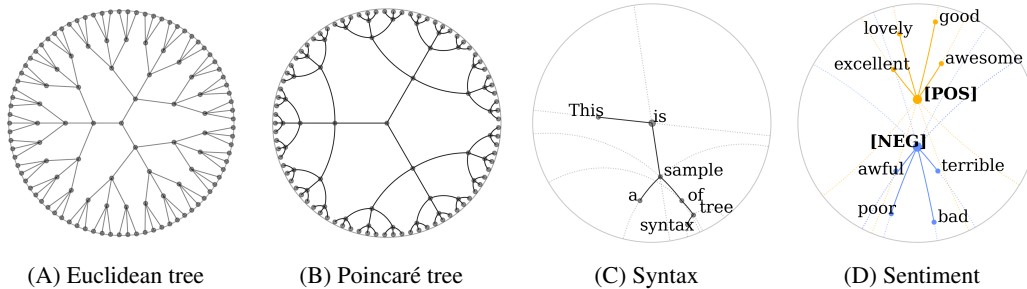

(A) Euclidean tree    (B) Poincaré tree    (C) Syntax    (D) Sentiment

Figure 1: Visualization of different spaces. (A, B) Comparison between trees embedded in Euclidean space and hyperbolic space. We use *geodesics*, the analogy of straight lines in hyperbolic spaces, to connect nodes in (B). Line/geodesic segments connecting nodes are approximately *of the same length* in their corresponding spaces. Intuitively, nodes embedded in Euclidean space look more "crowded", while the hyperbolic space allows sufficient capacity to embed trees and enough distances between leaf nodes. (C) A syntax tree embedded in a Poincaré ball. Hierarchy levels correspond to syntactical depths. The higher level a word is in a syntax tree, the closer it is to the origin. (D) Sentiment words embedded in a Poincaré ball. Hierarchy is defined as the sentiment polarity. We assume two meta [POS] and [NEG] embeddings at the highest level. Words with stronger sentiments are closer to their corresponding meta-embeddings.

Before going any further, it is crutial to differentiate a probe and supervised parser (Hall Maudslay et al., 2020), and ask what makes a good probe. Ideally, a probe should correctly recover syntactic information *intrinsically contained in the embeddings*, rather than *being a powerful parser* by itself. So it is important that the probe should have restricted modeling power but still be *sensitive enough* to the existence of syntax. For embeddings without strong syntax information (e.g., randomly initialized word embeddings), a probe should not aim to assign high parsing scores (because this would *overestimate* the existence of syntax), while a parser aims for high scores no matter how bad the input embeddings are. The quality of a probe is defined by its *sensitivity* to syntax.

Our work of probing BERT in hyperbolic spaces is exploratory. As opposed to the Euclidean syntactic subspaces in Hewitt & Manning (2019), we consider the Poincaré syntactic subspace, and show its effectiveness for recovering syntax. Figure 1 (C) gives an example of the reconstructed dependency tree embedded in the Poincaré ball. In our experiments, we highlight two important observations of our Poincaré probe: (a) it does *not* give higher parsing scores to baseline embeddings (which have no syntactic information) than Euclidean probes, meaning that it is *not a better parser*; (b) it reveals higher parsing scores, especially for *deeper trees, longer edges, and longer sentences*, than the Euclidean probe with strictly restricted capacity. Observation (b) can be interpreted from two perspectives: (1) it indicates that the Poincaré probe might be more *sensitive to the existence of deeper syntax*; (2) the structure of syntactic subspaces of BERT could be different than Euclidean, especially for deeper trees. Consequently, the Euclidean probe may *underestimate* the syntactic capability of BERT, and BERT may exhibit stronger modeling power for deeper syntax in some special metric space, in our case, a Poincaré ball.

To best exploit the inductive bias for hierarchical structures of hyperbolic space, we generalize our Poincaré probe to sentiment analysis. We construct a Poincaré sentiment subspace by predicting sentiments of individual words using vector geometry (Figure 1 D). We assume two *meta representations* for the positive and negative sentiments as the roots in the sentiment subspace. The stronger a word's polarity is, the closer it locates to its corresponding meta embedding. In our experiments, with clearly different geometric properties, the Poincaré probe shows that BERT encodes sentiments for each word in a very fine-grained way. We further reveal how the localization of word embeddings may change according to lexically-controlled contextualization, i.e., how different contexts would affect the geometric location of the embeddings in the sentiment subspace.

In summary, we present an Poincaré probe to reveal hierarchical linguistic structures encoded in BERT. From a hyperbolic deep learning perspective, our results indicate the possibility of using Poincaré models for learning better representations of linguistic hierarchies. From a linguistic perspective, we reveal the geometric properties of linguistic hierarchies encoded in BERT and posit that

BERT may encode linguistic information in special metric spaces that are not necessarily Euclidean. We demonstrate the effectiveness of our approach with extensive experiments and visualization.

## 2  RELATED WORK

**Probing BERT**     Recently, there are increasing interests in finding linguistic information encoded in BERT (Rogers et al., 2020). One typical line of work is the structural probes aiming to reveal how syntax trees are encoded geometrically in BERT embeddings (Hewitt & Manning, 2019; Reif et al., 2019). Our Poincaré probe generally follows this line and exploits the geometric properties of hyperbolic spaces for modeling trees. Again, we note that the goal of syntactic probes is to find syntax trees with strictly limited capacity, i.e., *a probe should not be a parser* (Hewitt & Manning, 2019; Kim et al., 2020), and strictly follow this restriction in our experiments. Other probing tasks consider a variety of linguistic properties, including morphology (Belinkov et al., 2017), word sense (Reif et al., 2019), phrases (Kim et al., 2020), semantic fragments (Richardson et al., 2020), and other aspects of syntax and semantics (Tenney et al., 2019). Our extended Poincaré probe for sentiment can be viewed as one typical semantic probe that reveals how BERT geometrically encodes word sentiments.

**Hyperbolic Deep Learning**     Recently, methods using hyperbolic geometry have been proposed for several NLP tasks due to its better inductive bias for capturing hierarchical information than Euclidean space. Poincaré embeddings (Nickel & Kiela, 2017) and POINCARÉGLOVE (Tifrea et al., 2019) learn embeddings of hierarchies using Poincaré models and exhibit impressive results, especially in low dimension. These works show the advantages of hyperbolic geometry for modeling trees while we focus on using hyperbolic spaces for probing contextualized embeddings. To learn models in hyperbolic spaces, previous works combine the formalism of Möbius gyrovector spaces with the Riemannian geometry, derive hyperbolic versions of important mathematical operations such as Möbius matrix-vector multiplication, and use them to build hyperbolic neural networks (Ganea et al., 2018). Riemannian adaptive optimization methods (Bonnabel, 2013; Bécigneul & Ganea, 2019) are proposed for gradient-based optimization. Techniques in these works are used as the infrastructure in this work for training Poincaré probes.

## 3  POINCARÉ PROBE

We begin by reviewing the basics of Hyperbolic Geometry. We follow the notations from Ganea et al. (2018). A generalized Poincaré ball is a typical model of hyperbolic space, denoted as $(\mathbb{D}_c^n, g_{\boldsymbol{x}}^{\mathbb{D}})$ for $c > 0$, where $\mathbb{D}_c^n = \{\boldsymbol{x} \in \mathbb{R}^n \mid c\|\boldsymbol{x}\|^2 < 1\}$ is a Riemannian manifold, $g_{\boldsymbol{x}}^{\mathbb{D}} = (\lambda_{\boldsymbol{x}}^c)^2 \boldsymbol{I}_n$ is the metric tensor, $\lambda_{\boldsymbol{x}}^c = 2/(1 - c\|\boldsymbol{x}\|^2)$ is the conformal factor and $c$ is the negative curvature of the hyperbolic space. We will use the term hyperbolic and Poincaré interchangeably according to the context. Our Poincaré probe uses the standard Poincaré ball $\mathbb{D}_c^n$ with $c = 1$. The distance function for $\boldsymbol{x}, \boldsymbol{y} \in \mathbb{D}_c^n$ is:

$$d_{\mathbb{D}}(\boldsymbol{x}, \boldsymbol{y}) = (2/\sqrt{c}) \tanh^{-1}(\sqrt{c}\| - \boldsymbol{x} \oplus_c \boldsymbol{y}\|), \tag{1}$$

where $\oplus_c$ denotes the *Möbius addition*, the hyperbolic version of the addition operator. Note that we recover the Euclidean space $\mathbb{R}^n$ when $c \to 0$. Additionally, we use $\boldsymbol{M} \otimes_c \boldsymbol{x}$ to denote the *Möbius matrix-vector multiplication* for a linear map $\boldsymbol{M} : \mathbb{R}^n \to \mathbb{R}^m$, which is the hyperbolic version of linear transforms. We use $\exp_{\boldsymbol{x}}^c(\cdot)$ to denote the *exponential map*, which maps vectors in the tangent space (in our case, a space projected from the BERT embedding space) to the hyperbolic space. Their closed-form formulas are detailed in Appendix C.

Our probes consist two simple linear maps $\boldsymbol{P}$ and $\boldsymbol{Q}$ that project BERT embeddings into a Poincaré syntactic/sentiment subspace. Formally, let $\mathcal{M}$ denote a pretrained language model that produces a sequence of distributed representations $\boldsymbol{h}_{1:t}$ given a sentence of $t$ words $\boldsymbol{w}_{1:t}$. We train a linear map $\boldsymbol{P} : \mathbb{R}^n \to \mathbb{R}^k$, $n$ being the dimension of contextualized embeddings and $k$ being the probe rank, that projects the distributed representations to the tangent space. Then the exponential map projects the tangent space to the hyperbolic space. In the hyperbolic space, we construct the Poincaré syntactic/sentiment subspace via another linear map $\boldsymbol{Q} : \mathbb{R}^k \to \mathbb{R}^k$. The equations are:

$$\boldsymbol{p}_i = \exp_{\boldsymbol{0}}(\boldsymbol{P} \boldsymbol{h}_i) \tag{2}$$
$$\boldsymbol{q}_i = \boldsymbol{Q} \otimes_c \boldsymbol{p}_i \tag{3}$$

Table 1: Results of tree distance and depth probes. We highlight two observations of Poincaré probes compared with Euclidean probes: (a) they do not assign higher scores to embeddings without syntactic information (ELMo0 and LINEAR), meaning that they do not form a parser; (b) they recover higher scores (colored cells) for contextualized embeddings with smaller probe capacity, meaning that they are more sensitive to the existence of syntax in contextualized embeddings.

| | Distance | | | | Depth | | | |
| | Euclidean | | Poincaré | | Euclidean | | Poincaré | |
| Method | UUAS | DSpr. | UUAS | DSpr. | Root % | NSpr. | Root % | NSpr. |
| --- | --- | --- | --- | --- | --- | --- | --- | --- |
| ELMo0 | 26.8 | 0.44 | 25.8 | 0.44 | 54.3 | 0.56 | 53.5 | 0.49 |
| LINEAR | 48.9 | 0.58 | 45.7 | 0.58 | 2.9 | 0.27 | 4.5 | 0.26 |
| ELMo1 | 77.0 | 0.83 | 79.8 | 0.87 | 86.5 | 0.87 | 88.4 | 0.87 |
| BERTBASE7 | 79.8 | 0.85 | 83.7 | 0.88 | 88.0 | 0.87 | 91.3 | 0.88 |
| BERTLARGE15 | 82.5 | 0.86 | 85.1 | 0.89 | 89.4 | 0.88 | 91.1 | 0.88 |
| BERTLARGE16 | 81.7 | 0.87 | **85.9** | **0.90** | 90.1 | **0.89** | **91.7** | **0.89** |

Here $P$ maps the original BERT embedding space to the tangent space of the origin of the Poincaré ball. Then $\exp_0(\cdot)$ maps the tangent space inside the Poincaré ball[2]. Consequently, in equation 3 we use the Möbius matrix-vector multiplication as the linear transformation in the hyperbolic space[3].

## 4 PROBING SYNTAX

Following Hewitt & Manning (2019), we aim to test if there exists a hyperbolic subspace transformed from the original BERT embedding space with simple parameterization where squared distances between embeddings or squared norms of embeddings approximate tree distances or node depths, respectively. The goal of the probe is to recover syntactic information *intrinsically contained in the embeddings*. To this end, a probe should *not* assign high parsing scores to baseline non-contextualized embeddings (otherwise it would become a *parser*, rather than being a *probe*). So it is crucial for the probe to have restricted modeling power (in our case, two linear transforms $P$ and $Q$) but still being *sensitive enough* for syntactic structures. We further test if the Poincaré probe is able to discover more syntactic information for deeper trees due to its intrinsic bias for modeling trees.

Similar to Hewitt & Manning (2019), we use the squared Poincaré distance to recreate tree distances between word pairs and the squared Poincaré distance to the origin to recreate the depth of a word:

$$\mathcal{L}_{\text{distance}} = \frac{1}{t^2} \sum_{i,j \in \{1,...,t\}} |d_{\text{T}}(w_i, w_j) - d_{\mathbb{D}^n}(\boldsymbol{q}_i, \boldsymbol{q}_j)^2| \tag{4}$$

$$\mathcal{L}_{\text{depth}} = \frac{1}{t} \sum_{i \in \{1,...,t\}} |d_{\text{D}}(w_i) - d_{\mathbb{D}^n}(\boldsymbol{q}_i, \boldsymbol{0})^2| \tag{5}$$

where $d_{\text{T}}(w_i, w_j)$ denotes the distance between word $i$, $j$ on their dependency tree, i.e., number of edges linking word $i$ to $j$ and $d_{\text{D}}(w_i)$ denotes the depth of word $i$ in the dependency tree. For optimization, we use the Adam (Kingma & Ba, 2014) initialized at learning rate 0.001 and train up to 40 epochs. We decay the learning rate and perform model selection based on the dev loss.

---

[2]The choice of tangent space at the origin, instead of other points, follows previous works (Ganea et al., 2018; Mathieu et al., 2019) for its mathematical simplexity and optimization convenience.

[3]This transformation is theoretically redundant, we use it primarily for numerical stability during optimization. We further note that such optimization stability is still an open problem in hyperbolic deep learning (Mathieu et al., 2019). We leave a detailed investigation to future work.

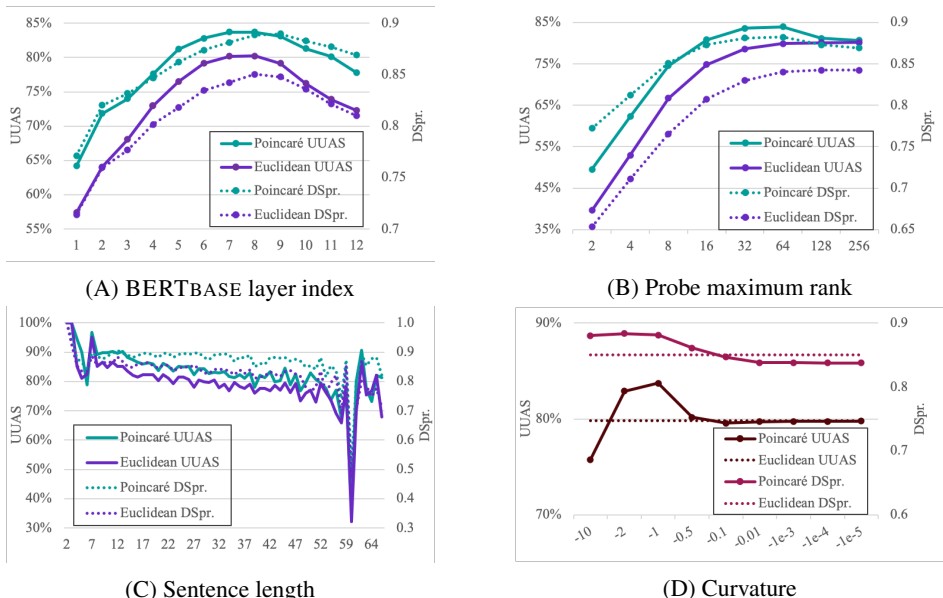

Figure 2: Comparison between the two probes. (A) Middle layered embeddings show richer syntactic information. (B) All probes recover syntax best at approximately rank 64 and Poincaré probes are especially better at low ranks. (C) Poincaré probes recover syntax better for longer sentences. (D) As the curvature goes closer to 0, Poincaré probes behave more similar to Euclidean probes.

## 4.1 Experimental Settings

Our experiments aim to demonstrate that the Poincaré probe better recovers deeper syntax in BERT without becoming a parser. We denote the probes in Hewitt & Manning (2019) Euclidean probes and follow their datasets and major baseline models. Specifically, we use the Penn Treebank dataset (Marcus & Marcinkiewicz, 1993) and reuse the data processing code in Hewitt & Manning (2019) to convert the data format to Stanford Dependency (de Marneffe et al., 2006). For baseline models, we use (a) ELMo0: strong character-level word embeddings with no contextual information. All probes should not find syntax trees from these embeddings. (b) LINEAR: left-to-right structured trees that only contain positional information. (c) ELMo1 and BERT*: strong contextualized embeddings with rich syntactic information. All probes should accurately recover all parse trees encoded in them.

Since the goal of probing is to recover syntax trees in a strict notion, we restrict our Poincaré probe to 64 dimension, i.e. $k = 64$ for $P$ and $Q$, which is at the same level, or smaller than the effective rank of the Euclidean probes reported in Hewitt & Manning (2019). We also emphasize that the parameters of our Poincaré probe are simply two matrices, which is again significantly less than a typical deep neural network parser (Dozat & Manning, 2016).

To evaluate the tree distance probes, we report: (a) undirected unlabeled attachment scores (UUAS), the scores showing if unlabeled edges are correctly recovered, against the gold undirected tree; (b) distance Spearman Correlation (DSpr), the scores showing how the recovered distances (either Euclidean or Hyperbolic) correlate with gold tree distances. To evaluate the tree depth probes, we report: (a) norm Spearman Correlation (NSpr), the scores showing how the true depth ordering correlates with the predicted depth ordering; (b) the correctly identified root nodes (root%), the scores showing to what extent the probes can identify sentence roots.

## 4.2 Results

Table 1 shows the major results for tree distance and depth probing. We see that Poincaré probes do *not* give higher scores than Euclidean probes for ELMo0 and LINEAR (i.e., they are not parsers. Also see appendix A.1 how a GRU transforms probes to be parsers), yet recovers higher parsing

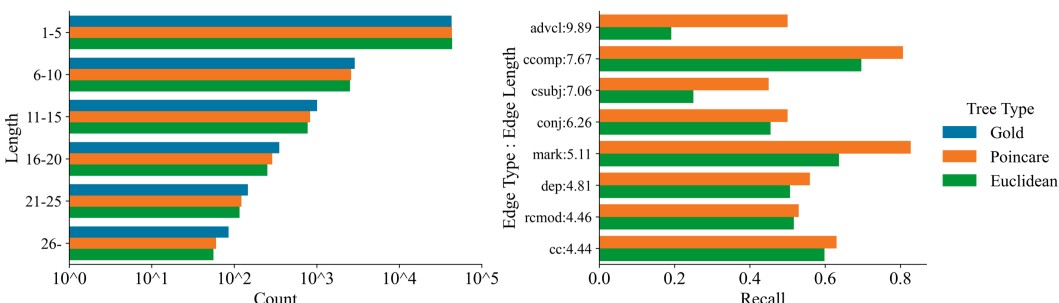

Figure 3: Left: comparison of edge length distributions. Distribution of the Poincaré probe aligns better with the ground truth than the Euclidean probe. Right: edge prediction recall of top longest edge types. The Poincaré probe is especially better at recovering edges of longer average length.

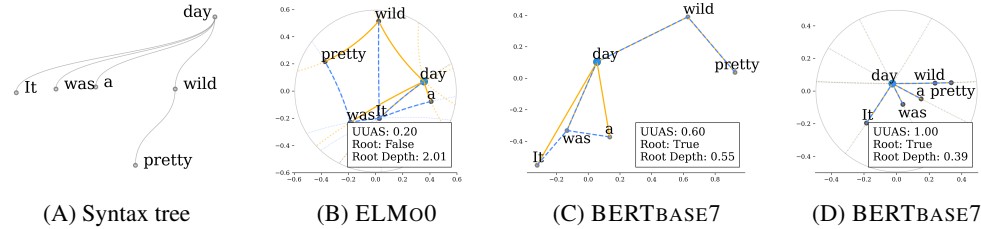

(A) Syntax tree      (B) ELMO0      (C) BERTBASE7      (D) BERTBASE7

Figure 4: PCA projection of dependency trees for the sentence *it was a pretty wild day*. Yellow lines/geodesics denote the ground truth and blue dashed lines/geodesics are predicted by the probe. Blue points denote root words of sentences. Word depths are clearly organized in the Poincaré ball (D) than the Euclidean space (C). The closer a word is to the origin, the upper level it is in the tree.

scores for embeddings with deep contextualization. The results indicate that: (a) Poincaré probes might be more sensitive to the existence of syntactic information; (b) the syntactic information encoded in contextualized embeddings may be *underestimated* by the Euclidean probes due to their incapacity of modeling trees. We further give a detailed analysis of the two probes from various perspectives on the distance task, which is harder than the depth task.

**Different BERTBASE Layers**     Figure 2A reports the distance scores of both the Poincaé probe and Euclidean probe trained on each layer of BERTBASE. The two are consistent with each other and show a similar tendency that syntax primarily exists in the middle layers.

**Probe Rank**     Figure 2B reports the distance scores of probes with different rank $k$ for BERTBASE7. Similar to Hewitt & Manning (2019), we see that the scores do not increase after 64 dimension with the Poincaré probe. We posit there might be some *intrinsic syntactic subspace* whose dimension is close to 64, but leave further investigation to future work.

**Sentence and Edge Lengths**     Figure 2C reports the distance scores of sentences with different lengths for BERTBASE7. The result that Poincaré probes give higher scores for longer sentences than Euclidean probes, meaning that they better reconstruct deeper syntax. Figure 3 (left) compares edge distance (length) distributions between ground truth edges and edges predicted by Euclidean and Poincaré probes, respectively. The distribution from the Poincaré probes are closer to the ground truth distribution, especially for longer trees. Figure 3 (right) shows that Poincaré probes consistently achieve better recall for edge types with longer average length. Figure 11 in the Appendix further compares minimum spanning tree results from predicted squared distances on BERTBASE7 (we randomly sample 12 instances from the dev set). These observations would be particularly interesting from a linguistic perspective as the syntactic structure for longer sentences are more complicated and challenging than those for shorter sentences. A promising future direction would be using hyperbolic spaces for parsing.

**Curvature of the Hyperbolic Space**     We further characterize the structure of the hyperbolic syntactic subspace with the curvature parameter, which measures how "curved" the space is (Figure 2D). We see that the optimal curvature is about -1, which is the curvature of a standard Poincaré

Table 2: Classification accuracy on Movie Review dataset. Both Euclidean and Poincaré give 48.4 (nearly random guess) to baseline LINEAR embeddings, meaning neither of them form a classifier.

| | BiLSTM | | BERTBASE9 | | BERTBASE10 | |
| | | | Euclidean | Poincaré | Euclidean | Poincaré |
|---|---|---|---|---|---|---|
| Accuracy | 79.7 | Trainable | 81.7 | **84.9** | 83.5 | 84.2 |
| | | Fixed | 78.4 (-3.3) | 84.2 (-0.7) | 79.1 (-4.4) | **84.5 (+0.3)** |

ball. Additionally, if we gradually change the curvature to 0, the space would be "less curved" and more similar to the Euclidean space (more "flat"). Consequently, the Poincaré scores converge to the Euclidean scores. When the curvature is 0, we recover the Euclidean probe.

**Visualization of Syntax Trees**     To illustrate the structural differences between Euclidean and Poincaré syntactic subspaces, we visualize the recovered dependency trees in Figure 4. To simultaneously visualize edges and tree depth, we jointly train the two probing objectives in equations 4 and 5 for the same probe. PCA projection[4] is then used to visualize the syntax trees, similar to Reif et al. (2019). As is shown in Figure 4, compared with the Euclidean probe, the Poincaré probe shows a more organized word hierarchy: the root word *day* takes the position at the origin and is surrounded by other words according to tree depth. We further note that embeddings of ELMo0 contains no syntactic information and the corresponding tree looks meaningless.

### 4.3    SUMMARY ON SYNTACTIC PROBE

The manifold of BERT embeddings could be complicated and exhibit special geometric properties. Our probe essentially serves as a well-defined, differentiable, surjective function that maps the BERT embedding space to a low dimensional, moderately curved Poincaré ball (rather than a Euclidean space), and consequently leads to better reconstruction of syntax trees. This indicates that the manifold of BERT syntax may be geometrically more similar to the Poincaré ball than a Euclidean space. And of course, we cannot conclude that the syntactic subspaces are indeed hyperbolic. Rather than being conclusive, we aim to explore alternative models for BERT syntax and reveal the underlying geometric structures. Such geometric properties of BERT are still far from well-understood and there are still many open problems worth studying.

## 5    PROBING SENTIMENT

Having validated the effectiveness of reconstructing syntax trees of our Poincaré probe, in this section, we generalize the probe to semantic hierarchies and focus on sentiment. We first recover a Poincaré sentiment subspace with two meta positive and negative embeddings taking the topmost hierarchy then identify word localizations in this subspace. We highlight that our probe reveals that BERT encodes fine-grained sentiments (positive, negative and *neutral*) for *each word*, even though the probe is trained with *sentence-level binary labels*. To further examine how the context of a sentence may affect its word embeddings, we perform a qualitative lexically-controlled contextualization, i.e., to change the sentiment of a sentence by carefully changing the word choice according to common linguistic rules, and visualize how the localization of embeddings changes accordingly.

We use the same probe architecture described in Section 3 to construct the sentiment subspace. Again, a probe is parameterized by two matrices $P$ and $Q$ that project a BERT embedding into a Poincaré ball. We adopt sentiment labels for sentences as our supervision and use the Movie Review dataset (Pang & Lee, 2005) with simple binary labels (positive and negative). Details of this dataset are in Appendix B. Given a sentence with $t$ words $w_{1:t}$, we project its BERT embeddings according to equation 2 and 3 and obtain $q_i \in \mathbb{D}^k$. To interpret the classification procedure in terms of vector geometry, we set two trainable meta representations for positive and negative labels as $c_{\text{pos}}, c_{\text{neg}} \in \mathbb{D}^k$. The logits for the two classes $l_{\text{pos}}, l_{\text{neg}}$ are obtained by summing over the Poincaré

---

[4]Although the use of PCA in hyperbolic spaces would lead to certain levels of distortion, we note that it is empirically effective for visualization. As there is no perfect analogy of PCA in hyperbolic spaces (Pennec et al., 2018), we leave the investigation of dimension reduction for hyperbolic spaces to future work.

Table 3: Top sentiment words recovered by Euclidean and Poincaré probes. Colored words align better with human intuition (orange for positive and blue for negative).

| | | |
|---|---|---|
| Euclidean | POS | latch, horne, testify, birth, opened, landau, cultivation, bern, willingly, visit, cub, carr, iced, meetings, awake, awakening, eddy, wryly, protective, fencing |
| | NEG | worthless, useless, frustrated, fee, inadequate, rejected, equipped, schedule, useful, outdated, discarded, equipment, pointless, sounded, weakened |
| Poincaré | POS | funky, lively, connects, merry, documented, vivid, dazzling, etched, infectious, relaxing, evenly, robust, wonderful, volumes, capturing, splendid, floats, sturdy |
| | NEG | inactive, stifled, dissatisfied, discarded, insignificant, insufficient, erratic, indifferent, fades, wasting, arrogance, robotic, stil, trails, poorly, inadequate |

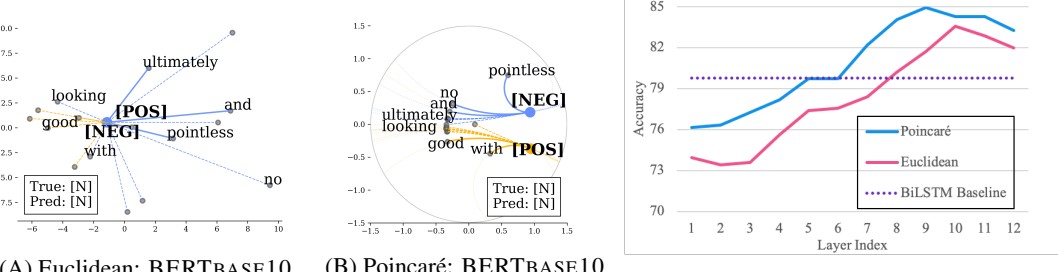

(A) Euclidean: BERTBASE10    (B) Poincaré: BERTBASE10    (C) BERTBASE layerwise accuracy

Figure 5: (A, B) PCA projection of sentence *a **good-looking** but **ultimately pointless** political thriller with plenty of action and almost **no** substance*. Words are connected to closer meta-embeddings. Words with dashed lines mean that the differences between their distances to two embeddings are not significant (neutral words). (C) Layerwise accuracy. Sentiment emerge at deeper layers (aroung layer 9) than syntax (around layer 7).

distance between each word and the opposite meta embeddings:

$$l_{\text{pos}} = \sum_{i=1}^{t} d_{\mathbb{D}^k}(\boldsymbol{q}_i, \boldsymbol{c}_{\text{neg}}) \qquad l_{\text{neg}} = \sum_{i=1}^{t} d_{\mathbb{D}^k}(\boldsymbol{q}_i, \boldsymbol{c}_{\text{pos}}) \qquad (6)$$

Since we know that the two classes are contrary to each other, we also consider assigning fixed positions for the two meta embeddings. For the Euclidean meta embeddings, we use $\boldsymbol{c}_{\text{pos}} = (1/\sqrt{k}) \cdot \mathbf{1}$ and $\boldsymbol{c}_{\text{neg}} = -\boldsymbol{c}_{\text{pos}}$ in the experiments, where $k$ is the space dimension. For the Poincaré meta embeddings, $\boldsymbol{c}_{\text{pos}} = \exp_{\mathbf{0}}((1/\sqrt{k}) \cdot \mathbf{1})$ and $\boldsymbol{c}_{\text{neg}} = -\boldsymbol{c}_{\text{pos}}$.

For training, we use cross-entropy after a Softmax as the loss function. An analog method is used for Euclidean Probes. We use RiemannianAdam (Bécigneul & Ganea, 2019) for all trainable parameters in the Poincaré ball, notably the two meta embeddings, and vanilla Adam for Euclidean parameters.

### 5.1 WORD POLARITIES BASED ON GEOMETRIC DISTANCES

Table 2 reports the classification accuracy. Firstly, both probes give better accuracy than a BiLSTM classifier, meaning that there exists rich sentiment information in contextualized embeddings. We note that when fixing the class meta representations, the Euclidean probe receives a large loss of performance, while the Poincaré probe can even perform better. This observation strongly suggests that the sentiment information may be encoded in some special geometric way. We also report the top sentiment words ranked by the distance gap between two meta-embeddings in Table 3 and see that the words recovered by the Poincaré probe align better with human intuition (colored words). More comprehensive word list are in Appendix B. Classification results for each BERT layer is shown in Figure 5C. Sentiment emerges at deeper layers (around 9) than syntax (around layer 7, comparing with Figure 2A), making it an evidence for the well-known assumption that syntax serves as the scaffold of semantics.

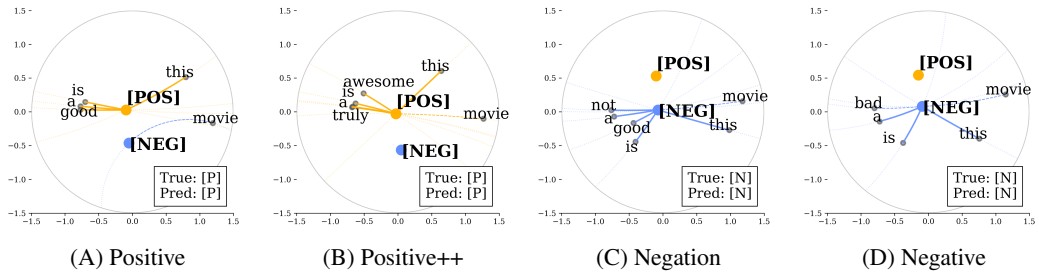

Figure 6: Lexically-controlled contextualization. (A) *This is a good movie*. (B) *This is a truly awesome movie*. (C) *This is not a good movie*. (D) *This is a bad movie*.

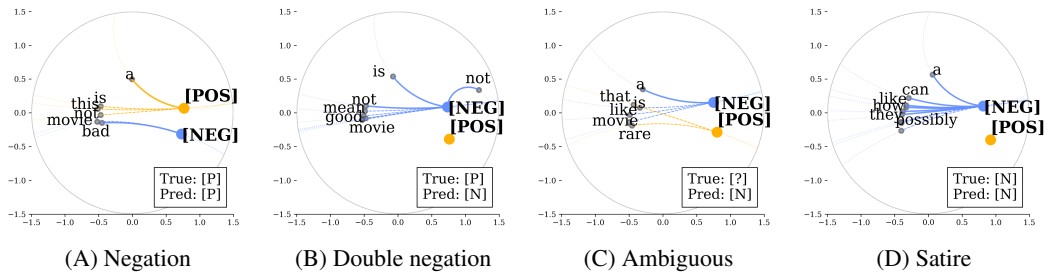

Figure 7: Special cases of lexically-controlled contextualization. (A) *This is not a bad movie*. (B) *I do not mean that movie is not good*. (C) *A movie like that is rare*. (D) *How can they possibly make a movie like that*.

## 5.2 VISUALIZATION AND LEXICALLY-CONTROLLED CONTEXTUALIZATION

Figure 5 illustrates how sentences are embedded in the two spaces. We see that: (a) both probes distinguish very fine-grained word sentiment, as one can infer if each word is positive, negative, or neutral , even if the probes are trained on sentence-level binary labels; (b) the Poincaré probe separates two meta-embeddings more clearly than the Euclidean probe and gives more compact embeddings. We emphasize that the observation in Figure 5 represents a general pattern, rather than being a special case. More visualization can be found in Appendix B.

To see how different contextualization may change the localization of embeddings, we carefully change the input words to control the sentiment. As is shown in Figure 6, we see that: (a) sentiment affects localization: stronger sentiments would result in closer distances to the meta-embeddings (Subfigure A v.s. B); (b) contextualization affects localization: when the sentence sentiment changes to negative (Subfigure A v.s. D), all words will be more close to the negative embedding, even when such change is induced by simple negation (Subfigure A v.s. C).

We further study more complicated cases to explore the limit of BERT embeddings (Figure 7) and find out: (a) BERT *fails at double negation* (Subfigure B), which shows that the logical reasoning behind double negation may be challenging for BERT; (b) BERT gives reasonable localization for an ambiguous sentence (Subfigure C) as most words do not significantly closer to one meta embeddings (dashed lines); (c) BERT gives correct localization for a sentence with satire (Subfigure D). More cases can be found in Appendix B. We further encourage the reader to run the Jupyter Notebook in supplementary materials to discover more visualization results.

## 6 CONCLUSION

In this paper, we present Poincaré probes that recover hyperbolic subspaces for hierarchical information encoded in BERT. Our exploration brings up new analytical tools and modeling possibilities about the geometry of BERT embeddings in hyperbolic spaces with detailed discussions about tree structures, localizations and their interactions with the contextualization.

ACKNOWLEDGMENTS

We thank all the reviewers for their valuable suggestions. This work is supported by Alibaba Group through Alibaba Research Intern Program.

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

APPENDIX

## A   ADDITIONAL RESULTS ON PROBING SYNTAX

Table 4 shows the score results of training Euclidean and Poincaré probes for distance and depth tasks simultaneously. More plots for longer sentences and deeper trees in the dev set are given (Figure 8 9 10) to illustrate the syntax subspaces (described in Section 3). (B) in these figures is produced by the $768d$ Euclidean Probe, and (C) (D) (E) are produced by the $64d$ Poincaré probe.

Table 4: Results of training probes for distance and depth tasks simultaneously.

| | Euclidean | | | | Hyperbolic | | | |
| | Distance | | Depth | | Distance | | Depth | |
| Method | UUAS | DSpr. | Root % | NSpr. | UUAS | DSpr. | Root % | NSpr. |
|---|---|---|---|---|---|---|---|---|
| ELMo0 | 20.1 | 0.41 | 55.0 | 0.51 | 19.3 | 0.41 | 56.3 | 0.53 |
| LINEAR | 43.4 | 0.57 | 8.6 | 0.27 | 45.1 | 0.57 | 7.0 | 0.27 |
| ELMo1 | 69.8 (-7.2) | 0.79 (-0.04) | 85.6 (-0.9) | 0.85 (-0.02) | 73.3 (-6.5) | 0.84 (-0.03) | 88.1 (-0.3) | 0.87 (-0.00) |
| BERTBASE7 | 73.8 (-6.0) | 0.81 (-0.04) | 83.3 (-4.7) | 0.84 (-0.03) | **78.2 (-5.5)** | **0.86 (-0.02)** | **88.6 (-2.7)** | **0.87 (-0.01)** |

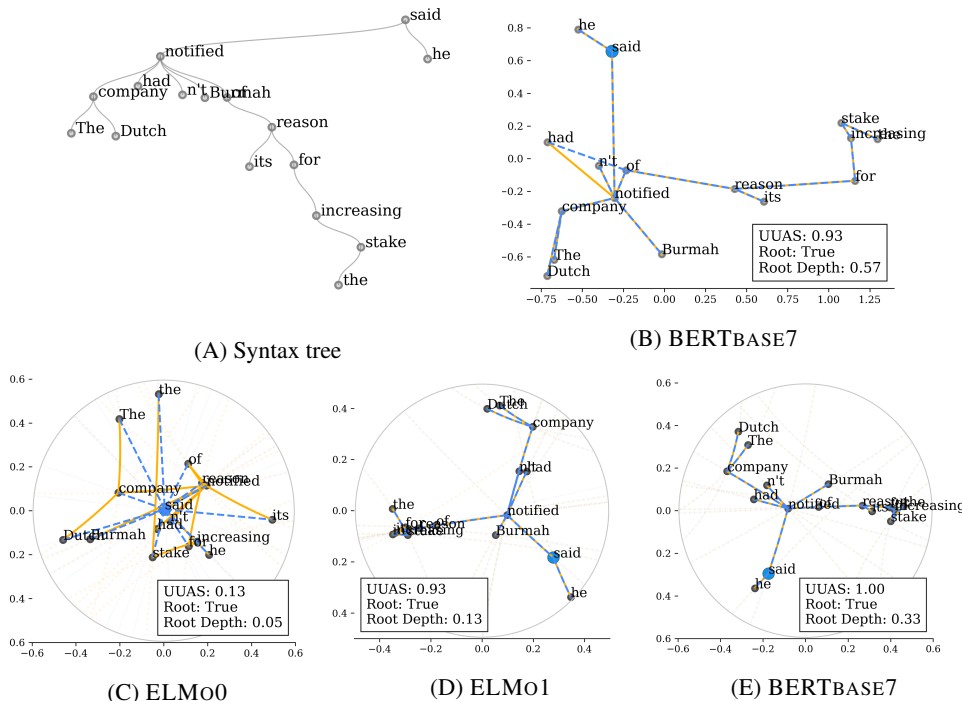

(A) Syntax tree    (B) BERTBASE7

(C) ELMo0    (D) ELMo1    (E) BERTBASE7

Figure 8: *The Dutch company hadn't notified Burmah of its reason for increasing the stake, he said.*

### A.1   A GRU LAYER TRANSFORMS A PROBE TO BE A PARSER

We show how contextualization transforms a probe to be a parser by adding a GRU layer to the probes. For Euclidean probe, hidden state of each word in GRU is used for encoding syntax, *i.e.*,

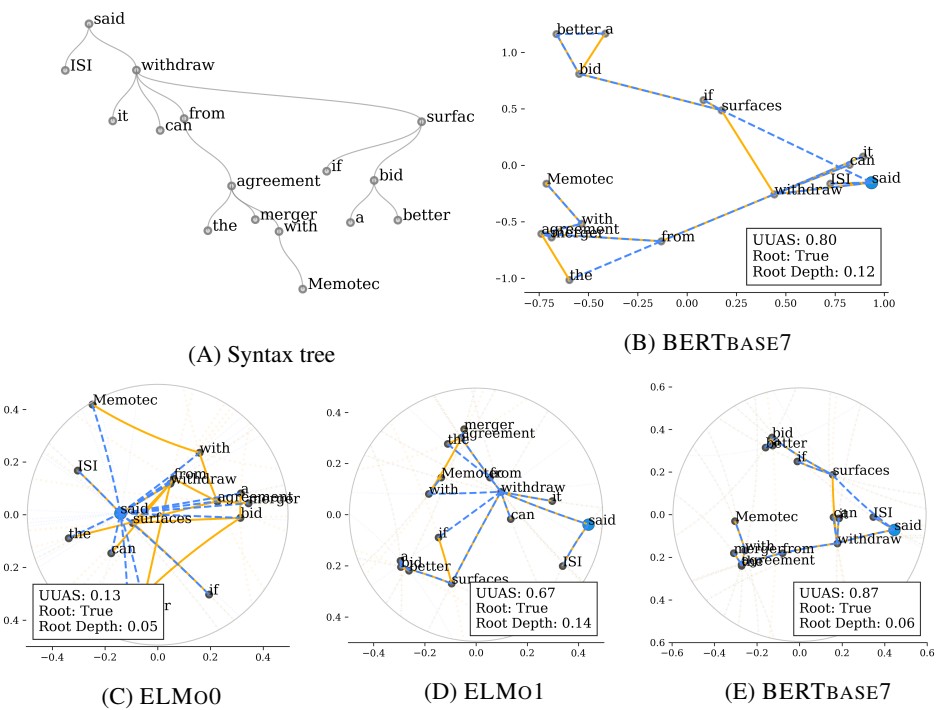

Figure 9: *ISI said it can withdraw from the merger agreement with Memotec if a better bid surfaces.*

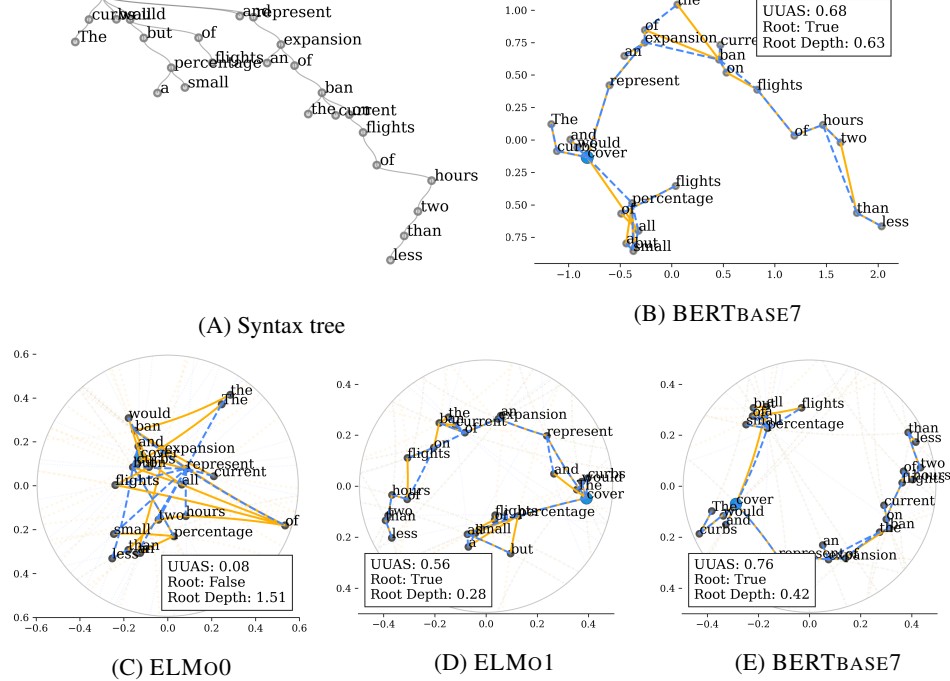

Figure 10: *The curbs would cover all but a small percentage of flights, and represent an expansion of the current ban on flights of less than two hours.*

$q_i = \text{GRU}(h_i)$. As for Poincaré probe, the equations are:

$$p_i = \exp_0(\text{GRU}(h_i)), \tag{7}$$
$$q_i = Q \otimes_c p_i. \tag{8}$$

Table 5 reports the scores, which are generally higher than probes based upon linear map. We note both of Poincaré probe and Euclidean probe obtain much higher scores on ELMo0, which indicates that these two probes can *learn* syntax trees, rather than probe from deep models.

Table 5: Results of training probes using local information for distance and depth tasks simultaneously.

| | Euclidean w/ GRU | | | | Hyperbolic w/ GRU | | | |
|---|---|---|---|---|---|---|---|---|
| | Distance | | Depth | | Distance | | Depth | |
| Method | UUAS | DSpr. | Root % | NSpr. | UUAS | DSpr. | Root % | NSpr. |
| ELMo0 | 68.4 | 0.81 | 73.3 | 0.82 | 74.1 | 0.84 | 86.4 | 0.86 |
| LINEAR | 46.7 | 0.57 | 7.8 | 0.27 | 46.3 | 0.58 | 9.4 | 0.27 |
| ELMo1 | **87.8** | **0.91** | 95.6 | **0.93** | 87.4 | **0.91** | **96.2** | **0.93** |
| BERTBASE7 | 79.3 | 0.87 | 93.0 | 0.92 | 86.8 | **0.91** | 96.0 | 0.92 |

## B ADDITIONAL RESULTS ON PROBING SENTIMENT

Movie Review [5] is a balanced sentiment analysis dataset with the same number of positive and negative sentences. Since there is no official split of this dataset, we randomly split 10% as dev and test set separately. The statistics can be found in Table 6.

Table 6: Statistics of Movie Review dataset.

| Dataset | #Train | #Dev | #Test | Avg. Len | Max Len |
|---|---|---|---|---|---|
| Movie Review | 8528 | 1067 | 1067 | 21 | 59 |

Table 8 is the extended version of Table 3. Since some neutral words can be equally close to the two meta embeddings, we reports top sentiment words ranked by the distance gap between two meta-embeddings. All subwords (begin with "##") and numbers are ignored.

More plots for sentences in the Movie Review dev set are given (Figure 12 13 14 15 16). Some neutral words with dashed lines are omitted for clarity. We note that in Figure 12, the positive part is more distinctly produced by Poincaré probe than Euclidean probe. In Figure 13, Poincaré probe correctly realizes "*but*" is turing to positive, but Euclidean Probe notes it as slightly negative, which is used in most cases. A similar pattern can be found in Figure 14. Figure 15 demonstrates the case that neither Poincaré probe nor Euclidean probe can correctly classify the sentence, as they may be disturbed by the name entity.

## C CLOSED-FORM FORMULAS OF MÖBIUS OPERATIONS

We restate the definitions of fundamental mathematical operations for the generalized Poincaré ball model. We refer readers to Ganea et al. (2018) for more details

**Möbius Addition** The *Möbius addition* for $x, y \in \mathbb{D}_c^n$ is defined as:

$$x \oplus_c y := \frac{(1 + 2c\langle x, y \rangle + c\|y\|^2)x + (1 - c\|x\|^2)y}{1 + 2c\langle x, y \rangle + c^2\|x\|^2\|y\|^2}. \tag{9}$$

---
[5] https://www.cs.cornell.edu/people/pabo/movie-review-data/

Table 7: Accuracy for probing on GLOVE and LINEAR.

|  | Euclidean | Poincaré |
|---|---|---|
| LINEAR | 48.4 | 48.4 |
| GLOVE | 75.9 | 76.7 |

Table 8: Top 100 sentiment words by Euclidean and Poincaré probes.

| | | |
|---|---|---|
| Euclidean | POS | latch, horne, testify, birth, opened, landau, cultivation, bern, willingly, visit, cub, carr, iced, meetings, awake, awakening, eddy, wryly, protective, fencing, clears, bail, levy, shy, doyle, belong, grips, regis, flames, initiation, macdonald, woolf, concludes, hostages, coco, elf, battista, darkly, intimate, closure, caine, ridley, beaches, pol, possession, paranormal, enhancing, gains, playground, bonds, alfonso, prom, pry, wonderland, cozy, warmth, axel, alex, duval, seal, cass, open, versa, openly, benign, stirred, parental, deeply, babies, fiercely, nick, ecstasy, lara, ri, alexandre, fiery, serra, interiors, ile, intimacy, claw, jacob, merry, secretly, eerie, maternal, kidnapping, transitions, jacques, claude, hierarchy, vibrant, warmly, illumination, benoit, wolf, fed, operative, interior, interference |
| | NEG | worthless, useless, frustrated, fee, inadequate, rejected, equipped, schedule, useful, outdated, discarded, equipment, pointless, sounded, weakened, undeveloped, received, ought, heaviest, conceived, recycled, hates, inactive, improper, muttering, pathetic, gee, unnecessary, humiliated, insignificant, timed, dated, sacrificed, bucks, drowned, ineffective, tired, compiled, wasted, hurried, exaggerated, needed, rational, hired, magazine, eighties, junk, biased, garbage, shall, weakly, redundant, flopped, insulting, thou, lifeless, interchange, flatly, hampered, poorly, inability, styled, solution, lame, indifferent, failed, rubbish, generic, accumulated, packaged, truncated, defeating, failure, trails, month, stale, deploy, could, bland, died, gymnastics, cared, graveyard, obsolete, unsuccessful, baked, suited, scripts, weathered, worst, amateur, disappointing, labelled, lazy, verse, chair, equation, directions, preliminary, calculated |
| Poincaré | POS | funky, lively, connects, merry, documented, vivid, dazzling, etched, infectious, relaxing, evenly, robust, wonderful, volumes, capturing, splendid, floats, sturdy, immensely, potent, reflective, graphical, energetic, inviting, illuminated, vibrant, delightful, illuminating, rewarded, stirring, charting, bursting, absorbing, enhancing, admired, recreated, enjoyable, stellar, refreshing, richly, swung, woven, demonstration, captures, sterling, guarantees, glorious, emblem, reflected, tidy, encouraging, stunning, beautifully, reward, flood, gorgeous, charming, fiery, feast, radiant, clears, powerful, searing, shimmering, compassionate, brave, warm, finely, exquisite, successfully, eminent, superb, playful, effortlessly, fiercely, neatly, evan, witty, confirming, candi, teasing, sincerely, excellent, piercing, lyrical, entertaining, reminder, enriched, beautiful, luminous, tremendous, engaging, arresting, irresistible, illustrates, fascinating, bracing, pleasant, exceptional, revive |
| | NEG | inactive, stifled, dissatisfied, discarded, insignificant, insufficient, erratic, indifferent, fades, wasting, arrogance, robotic, stil, trails, poorly, inadequate, disappointment, inferior, dissipated, useless, meaningless, fails, pathetic, lifeless, threw, flies, meek, flopped, intimidated, tainted, redundant, fee, failing, drained, prevents, hopeless, bland, disappointing, ruse, weighs, lacking, squash, hampered, fail, lazy, weaker, dripping, eroded, lame, neglect, failure, worthless, limp, nowhere, undermine, weighted, lax, washed, hollow, flaw, failed, weak, ruined, inappropriate, boring, inability, mud, miserable, disgusted, uneven, rebellious, sloppy, tired, lacks, defeating, weakly, coarse, relegated, baked, collapses, stranded, dissolve, dull, pointless, unnecessary, rejected, scarcely, drain, ted, slug, idiots, unsuccessful, dazed, thinner, exhausted, clumsy, ill, tires, losing, foolish |

**Möbius Matrix-vector Multiplication** For a linear map $\boldsymbol{M} : \mathbb{R}^n \to \mathbb{R}^m$ and $\forall \boldsymbol{x} \in \mathbb{D}_c^n$, if $\boldsymbol{M}\boldsymbol{x} \neq \boldsymbol{0}$, then the *Möbius matrix-vector multiplication* is defined as:

$$\boldsymbol{M} \otimes_c \boldsymbol{x} = (1/\sqrt{c}) \tanh \left( \frac{\|\boldsymbol{M}\boldsymbol{x}\|}{\|\boldsymbol{x}\|} \tanh^{-1}(\|\sqrt{c}\boldsymbol{x}\|) \right) \frac{\boldsymbol{M}\boldsymbol{x}}{\|\boldsymbol{M}\boldsymbol{x}\|}, \tag{10}$$

and $\boldsymbol{M} \otimes_c \boldsymbol{x} = \boldsymbol{0}$ if $\boldsymbol{M}\boldsymbol{x} = \boldsymbol{0}$.

**Exponential and Logarithmic Maps**    Let $T_{\boldsymbol{x}}\mathbb{D}_c^n$ denote the *tangent space* of $\mathbb{D}_c^n$ at $\boldsymbol{x}$. The *exponential map* $\exp_{\boldsymbol{x}}^c(\cdot) : T_{\boldsymbol{x}}\mathbb{D}_c^n \to \mathbb{D}_c^n$ for $\boldsymbol{v} \neq \mathbf{0}$ is defined as:

$$\exp_{\boldsymbol{x}}^c(\boldsymbol{v}) = \boldsymbol{x} \oplus_c \left( \tanh\left( \sqrt{c}\frac{\lambda_{\boldsymbol{x}}^c \|\boldsymbol{v}\|}{2} \right) \frac{\boldsymbol{v}}{\sqrt{c}\|\boldsymbol{v}\|} \right). \tag{11}$$

As the inverse of $\exp_{\boldsymbol{x}}^c(\cdot)$, the *logarithmic map* $\log_{\boldsymbol{x}}^c(\cdot) : \mathbb{D}_c^n \to T_{\boldsymbol{x}}\mathbb{D}_c^n$ for $\boldsymbol{y} \neq \boldsymbol{x}$ is defined as:

$$\log_{\boldsymbol{x}}^c(\boldsymbol{y}) = \frac{2}{\sqrt{c}\lambda_{\boldsymbol{x}}^c} \tanh^{-1}(\sqrt{c}\| - \boldsymbol{x} \oplus_c \boldsymbol{y}\|) \frac{-\boldsymbol{x} \oplus_c \boldsymbol{y}}{\| - \boldsymbol{x} \oplus_c \boldsymbol{y}\|}. \tag{12}$$

# D   ADDITIONAL RESULTS ON EUCLIDEAN PROBE VARIATIONS

We extend Euclidean probes to two linear transforms with non-linearity in between, which would make it more fair comparing to Poincaré Probe. The additional results are reported in Table 9.

Table 9: Results of training Euclidean probe variations for distance task.

| Method | | Euclidean | | | | Poincaré |
|---|---|---|---|---|---|---|
| | | No non-linearity | ReLU | Sigmoid | Tanh | |
| RANDOM | UUAS | 18.7 | 21.5 | 22.0 | 22.2 | 19.9 |
| | DSpr. | 0.39 | 0.41 | 0.40 | 0.41 | 0.40 |
| ELMO0 | UUAS | 26.7 | 29.5 | 29.8 | 29.6 | 25.8 |
| | DSpr. | 0.44 | 0.45 | 0.45 | 0.45 | 0.44 |
| LINEAR | UUAS | 48.3 | 48.5 | 47.8 | 48.2 | 45.7 |
| | DSpr. | 0.57 | 0.58 | 0.57 | 0.57 | 0.58 |
| BERTBASE7 | UUAS | 79.9 | 84.0 | 84.5 | 84.0 | 83.7 |
| | DSpr. | 0.84 | 0.88 | 0.88 | 0.88 | 0.88 |

# E   ADDITIONAL RESULTS ON CURVATURE

To visualize the embedded trees directly in $2d$ hyperbolic spaces, we train $2d$ probes with different curvatures for both distance and depth tasks simultaneously on BERTBASE7. The scores are reported in Table 10. The circles in (B), (C), (D) of Figure 17, 18, 19 show the boundary of the Poincaré models.

Table 10: Results of $2d$ probes with different curvatures for both distance and depth tasks on BERTBASE7.

| | | Distance | | Depth | |
|---|---|---|---|---|---|
| | Curvature | UUAS | DSpr | Root % | NSpr. |
| Euclidean | 0 | 24.3 | 0.51 | 77.6 | 0.80 |
| | -0.1 | 38.7 | 0.68 | 80.6 | 0.83 |
| Poincaré | -0.5 | 39.1 | 0.69 | 80.8 | 0.83 |
| | -1 | **40.5** | **0.70** | **81.5** | 0.83 |

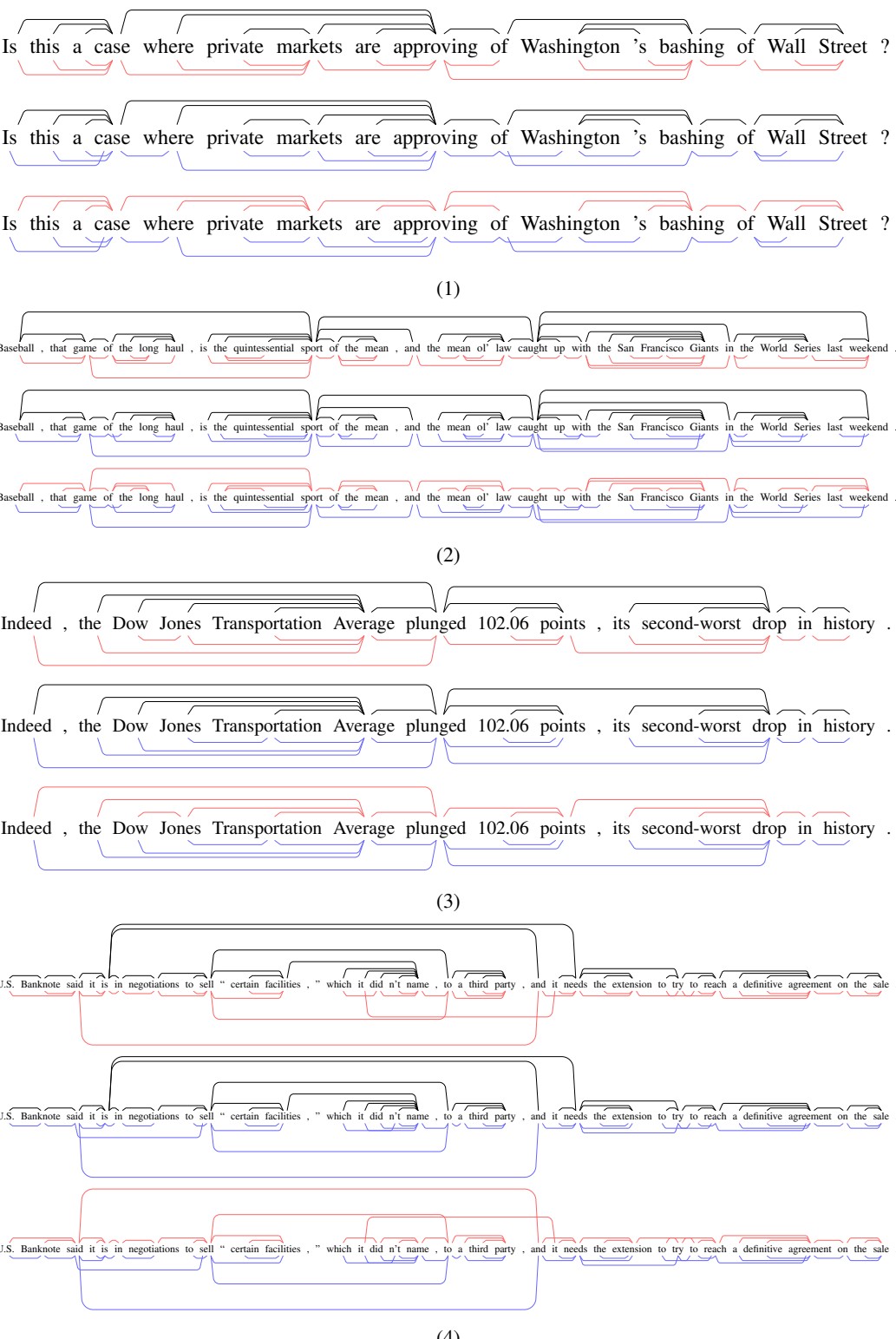

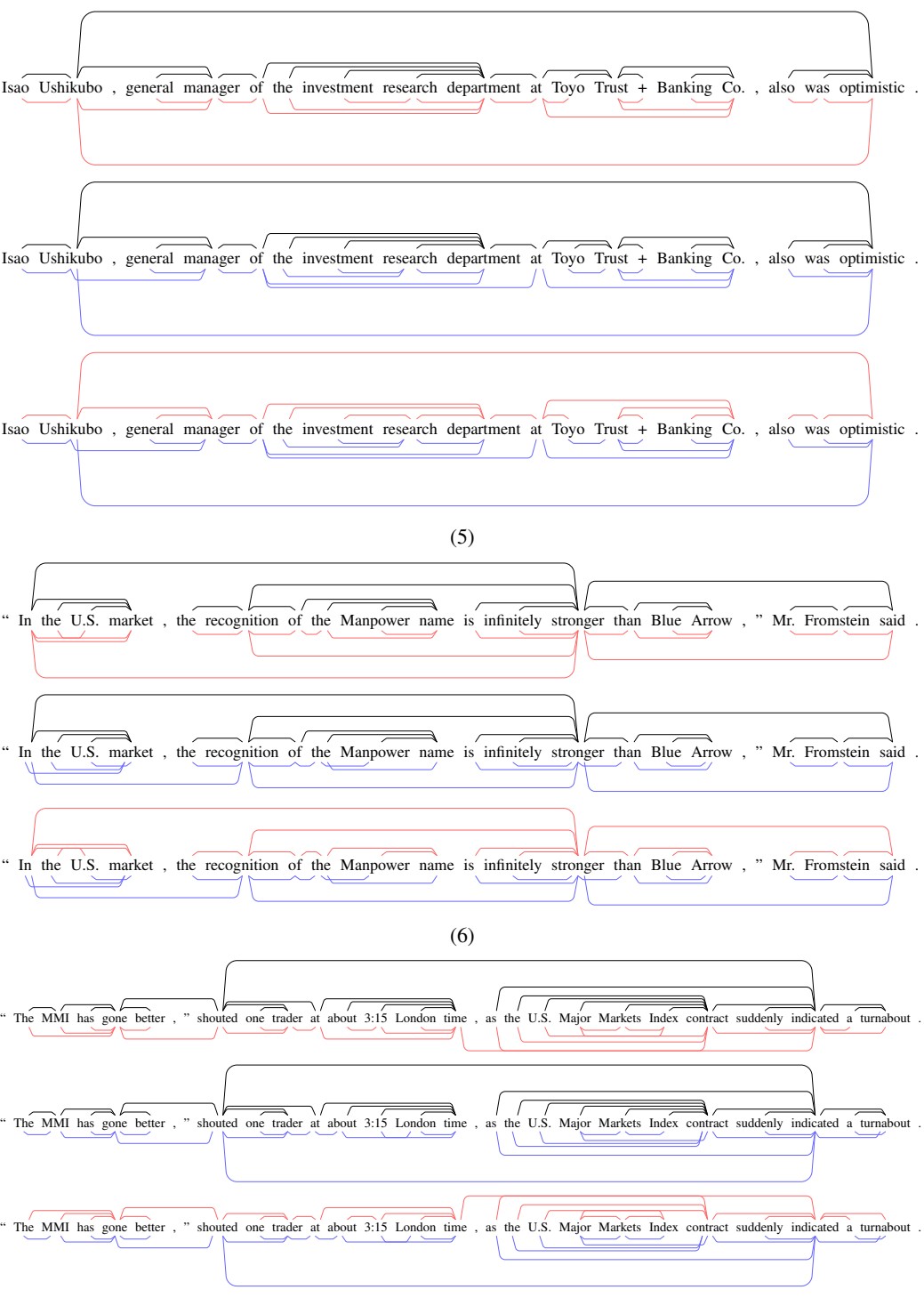

(5)

(6)

(7)

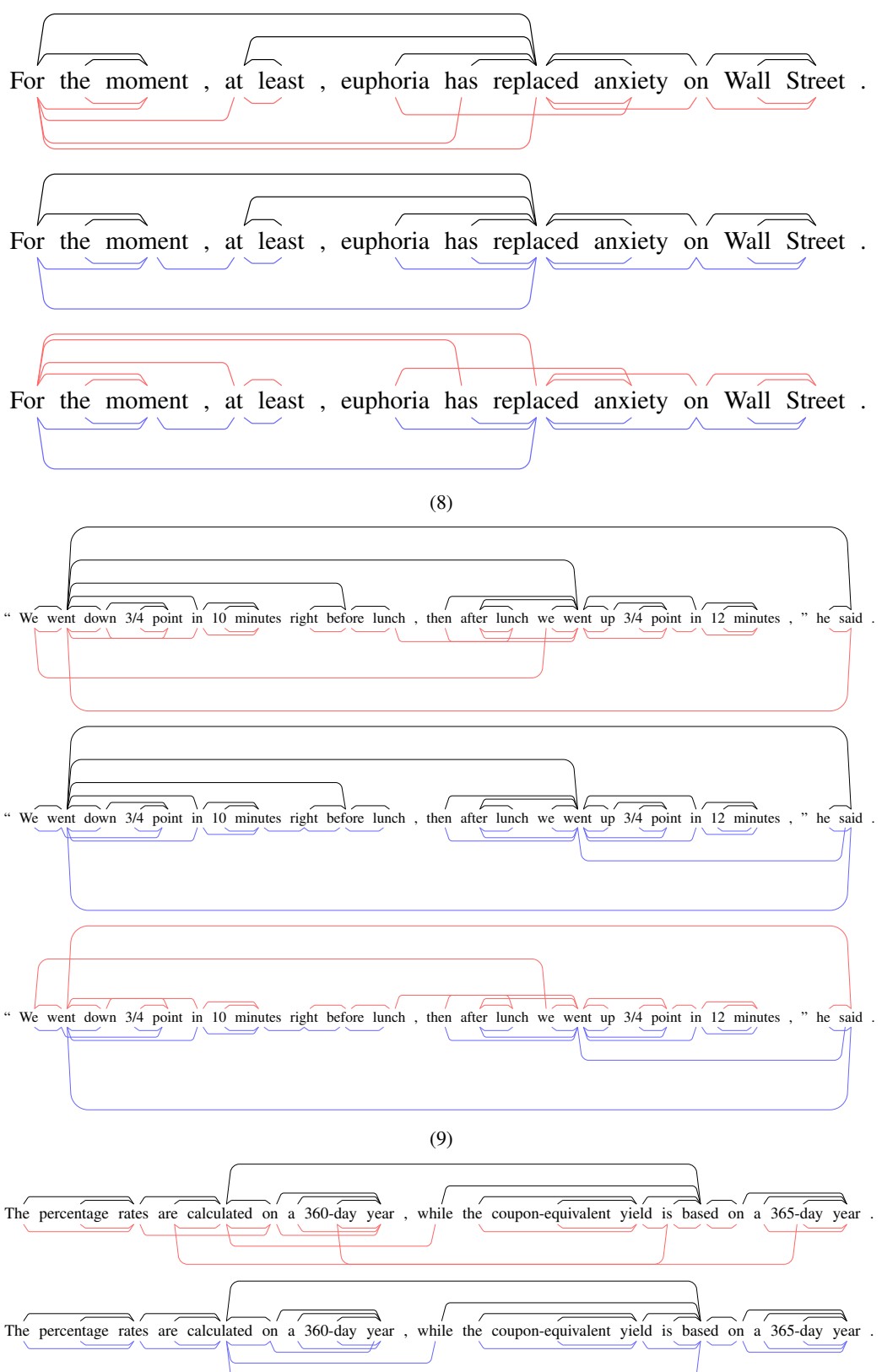

(8)

(9)

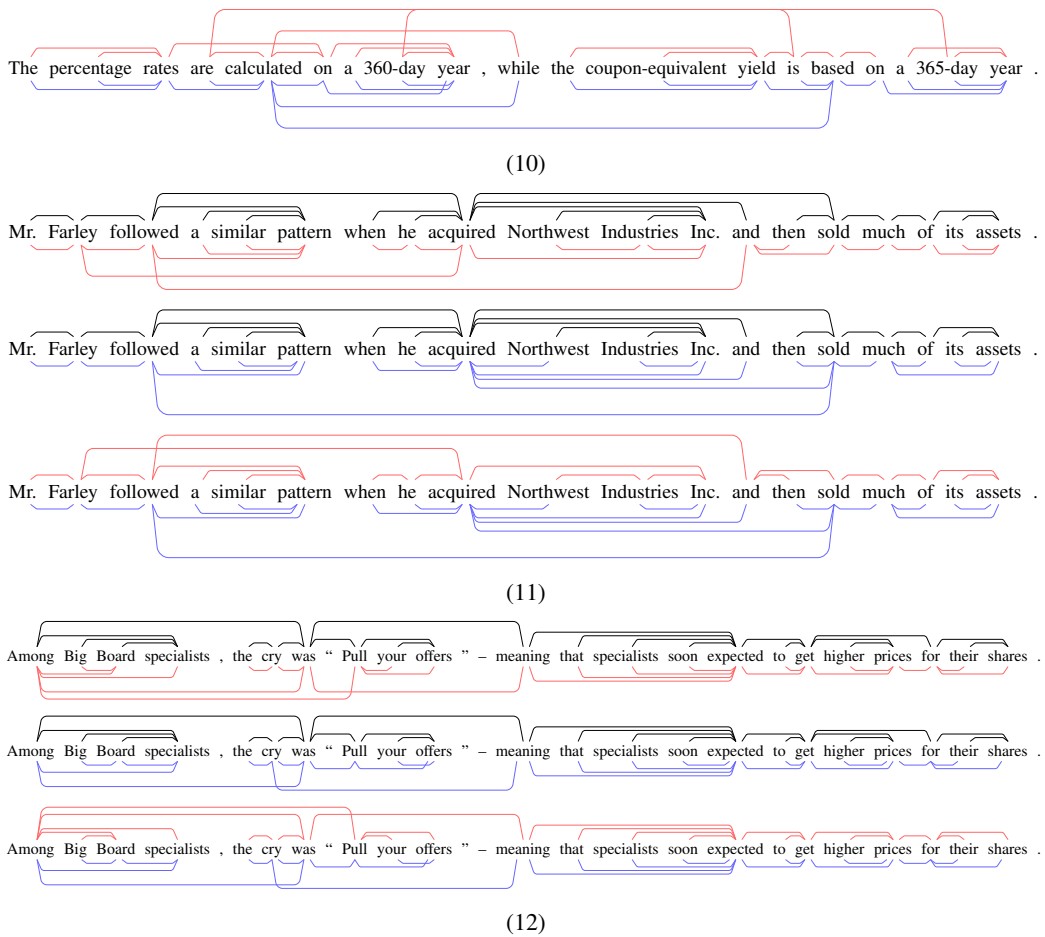

(10)

(11)

(12)

Figure 11: Minimum spanning trees resultant from predicted squared distances on BERTBASE7. Black edges are the gold parse, red edges are predicted by Euclidean probes, blue edges are predicted by Poincaré probes. The two probes primarily differs at long edges.

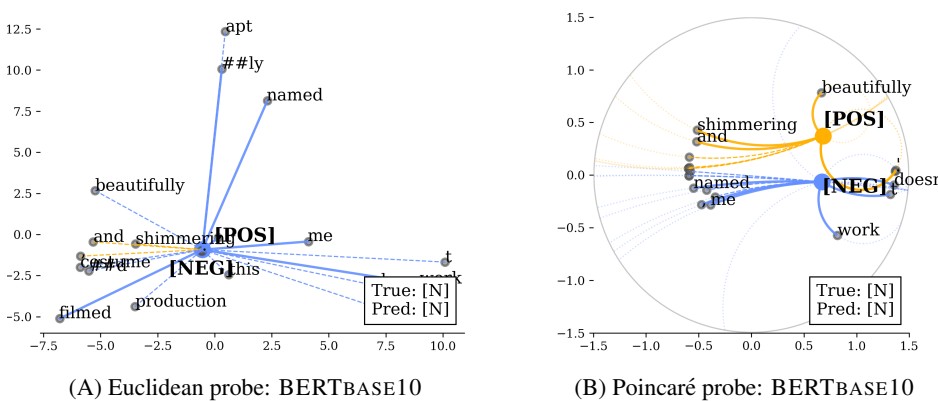

(A) Euclidean probe: BERTBASE10

(B) Poincaré probe: BERTBASE10

Figure 12: *Aptly named, this shimmering, beautifully costumed and filmed production doesn't work for me.*

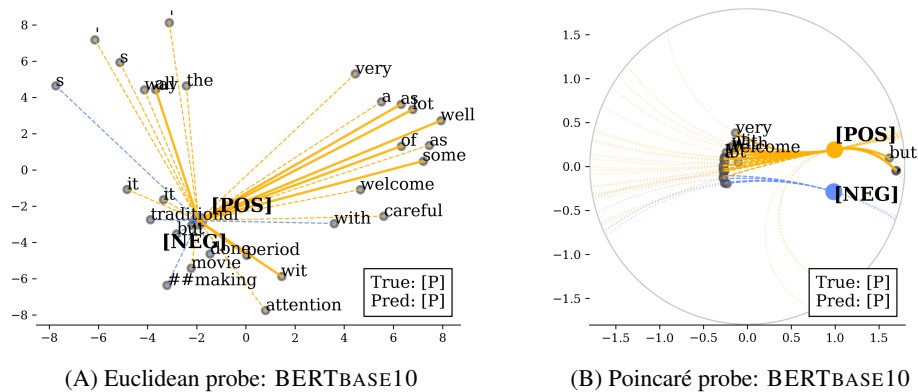

(A) Euclidean probe: BERTBASE10

(B) Poincaré probe: BERTBASE10

Figure 13: *It's traditional moviemaking all the way, but it's done with a lot of careful period attention as well as some very welcome wit.*

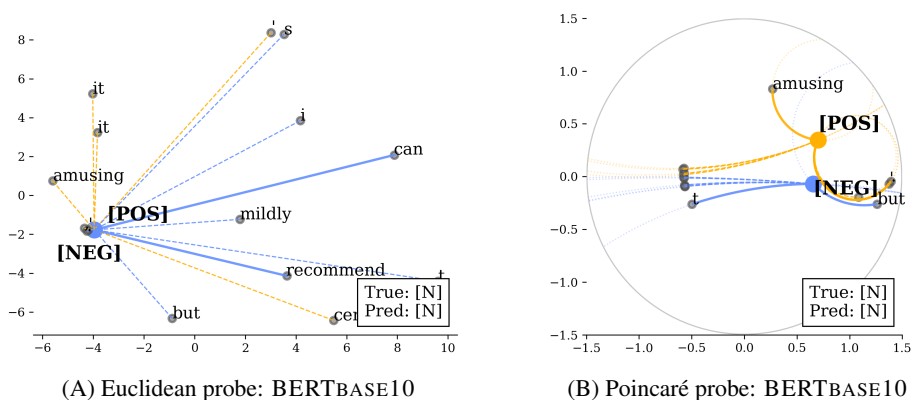

(A) Euclidean probe: BERTBASE10

(B) Poincaré probe: BERTBASE10

Figure 14: *It's mildly amusing, but I certainly can't recommend it.*

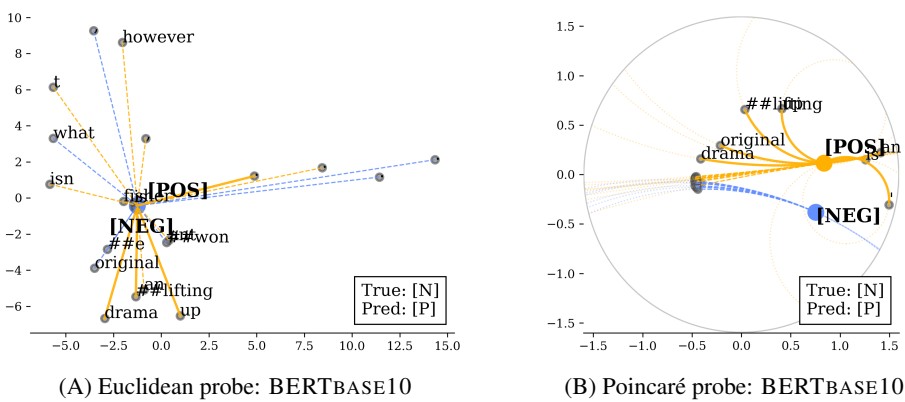

(A) Euclidean probe: BERTBASE10

(B) Poincaré probe: BERTBASE10

Figure 15: *An uplifting drama...what Antwone Fisher isn't, however, is original.*

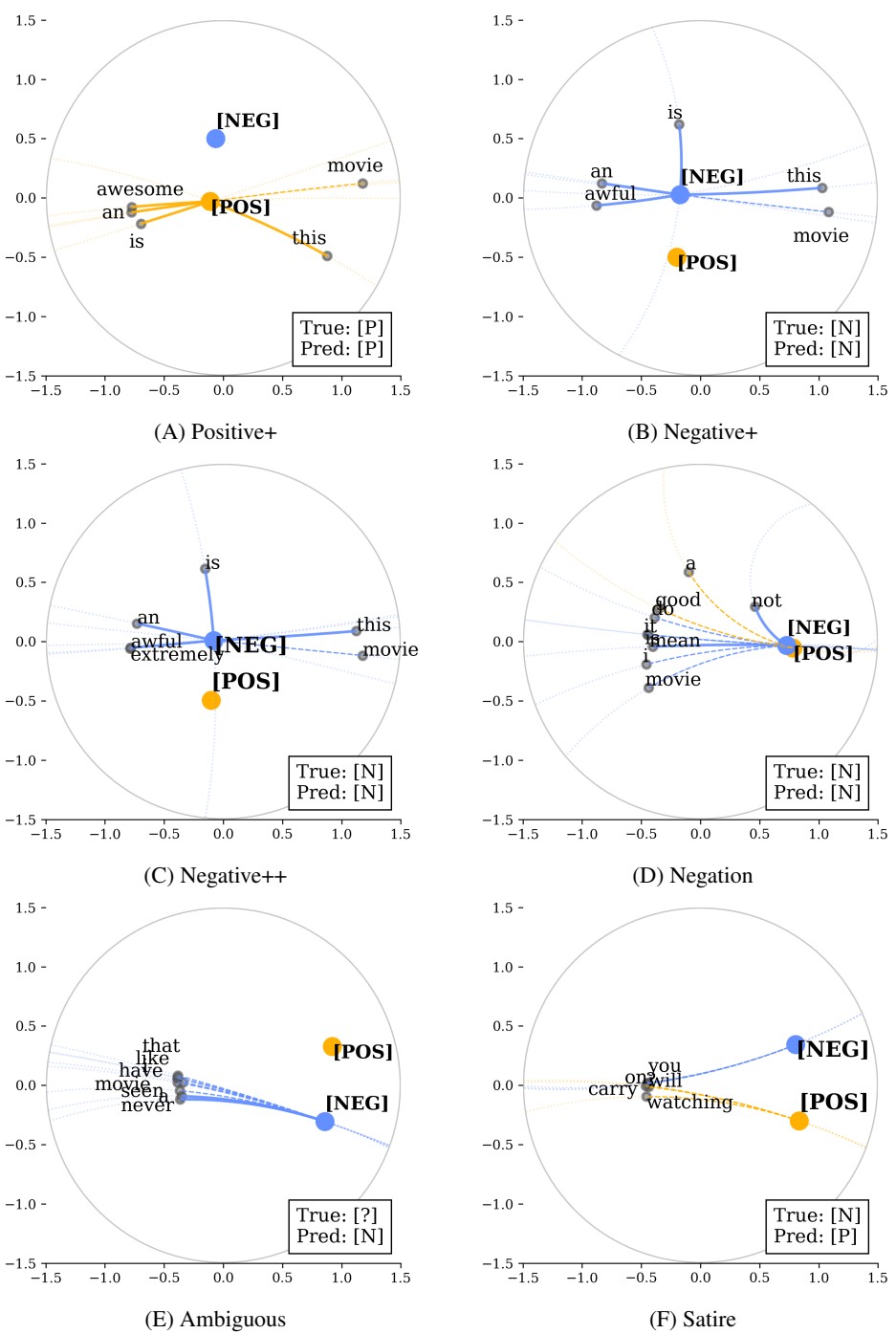

Figure 16: Additional special cases of lexically-controlled contextualization. (A) *This is an awesome movie.* (B) *This is an awful movie.* (C) *This is an extremely awful movie.* (D) *I do not mean it is a good movie.* (E) *I have never seen a movie like that.* (F) *You will carry on watching?*

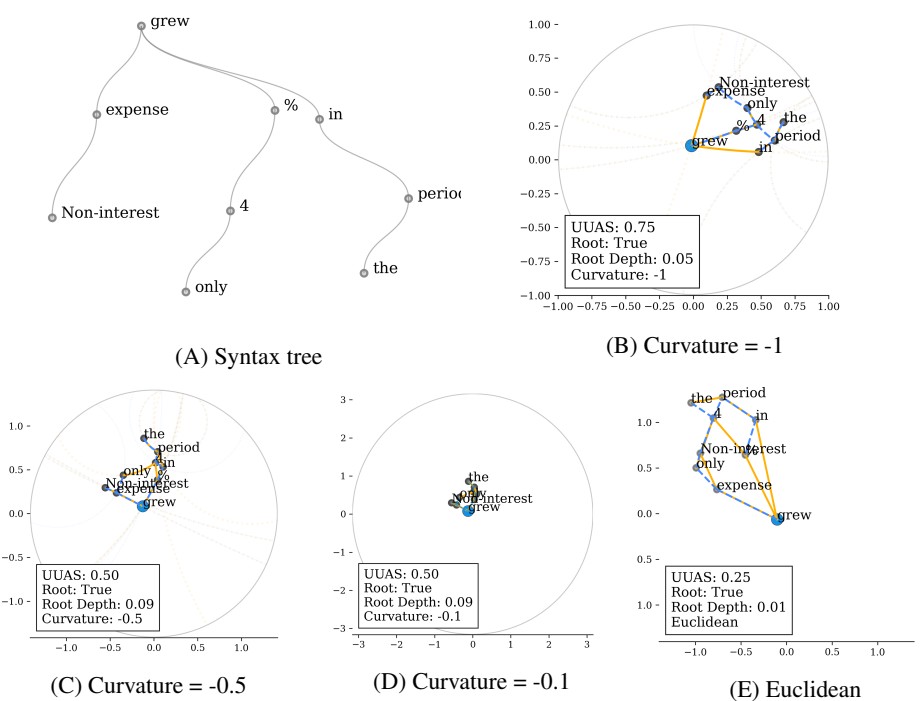

Figure 17: *Non-interest expense grew only 4% in the period.*

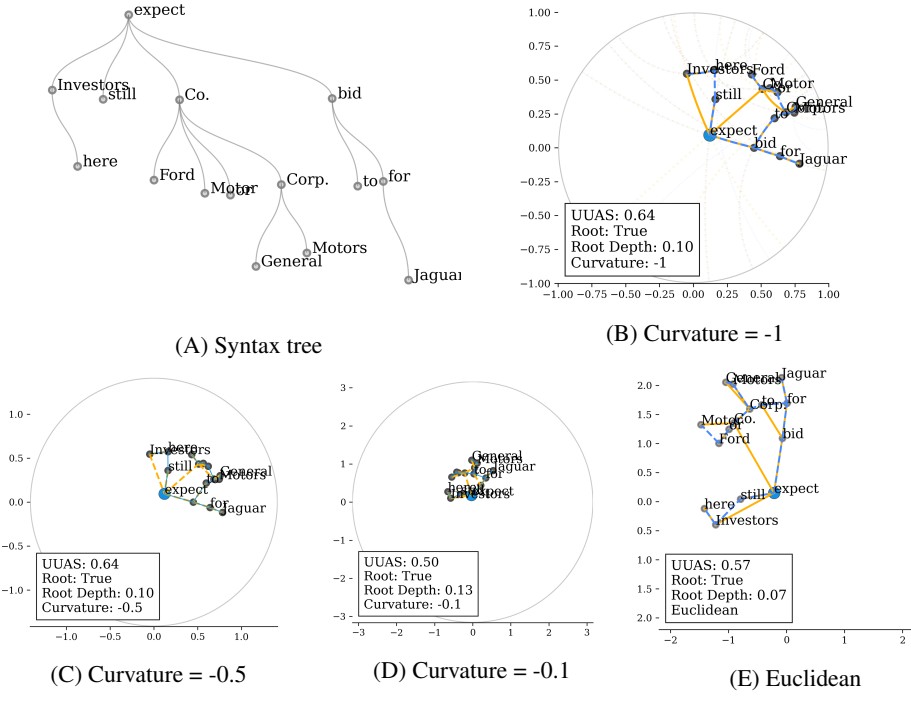

Figure 18: *Investors here still expect Ford Motor Co. or General Motors Corp. to bid for Jaguar.*

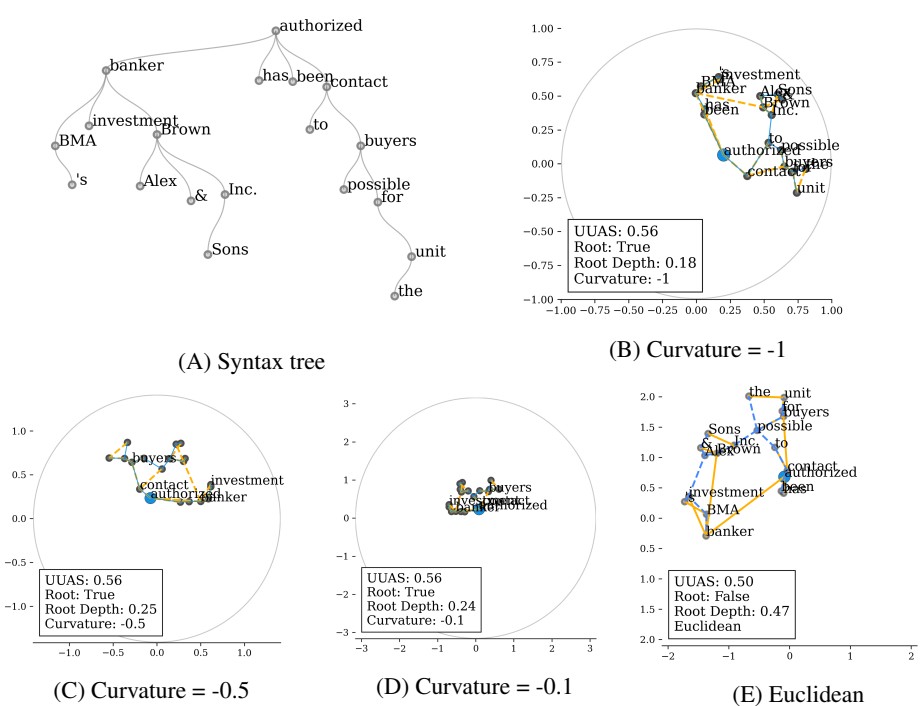

Figure 19: *BMA's investment banker, Alex. Brown & Sons Inc., has been authorized to contact possible buyers for the unit.*

