# OpenReview forum: "Probing BERT in Hyperbolic Spaces"
_ICLR.cc/2021/Conference — ICLR 2021 Poster_

### Official Review · AnonReviewer3 · 2020-10-21
**Official Blind Review #3 - edited after author response**

**Rating:** 6
**Confidence:** 3

**Review:**

This paper proposes probes based on hyperbolic embedding spaces, and compares them to the behaviour of Euclidean probes from recent work. The main result is that these probes allow for better recovery of syntactic properties of sentences from contextualized word embeddings compared to context-independent ones, when comparing them to euclidean probes. Similar results are presented  on sentiment analysis, even though no results are presented for context-independent word embeddings.

On the whole the paper is well-written with great visualizations. My main concern is that the findings are not strong enough at this point. As per the introduction, the first main finding is that the results indicate the possibility that hyperbolic models help us construct more sensitive probes. But how do we know a probe is not too sensitive or not sensitive enough? E.g. looking at the results in table 1, how do we know that the differences in the scores when using contextualised embeddings are for a good reason? Some of the differences are small, especially in the case of the depth probe. Similarly, for the second finding about BERT might be encoding information in non-Euclidean way, how do we know this is the case? One way would be to somehow modify BERT and make it "more Euclidean", and then this would help strengthen the hypothesis. But now as it stands, it is interesting but rather speculative.

Some other points:
- Statistical significance would help in reporting the results as many differences are small in Table 1. Also confidence intervals in the sentence length experiment would be nice.
- It would be better to explain the scores used in Table 1, instead of referring to Hewitt and Manning
- For syntactic parsing it is argued that the probe shouldn't be a parser, but in the sentiment analysis the probe is a better parser than the baseline model used. Shouldn't the probe not be a good sentiment analysis model in itself?
- In the ends of section 5, the referring labels for subfigures in figure 6 are off.

Post-author response: I appreciate the extra experiment on GLoVe vs Linear on sentiment analysis, it is what I was asking for. I have raised my score in response to that, as well as the additional reporting on the results. The discussion in Bayesian terms was interesting, I think it would help. Nevertheless, I was thinking is it that it should be possible to construct embeddings that have known syntactic vs semantic properties. E.g. one could increase/decrease the context size, perhaps to extreme values in the case of models such as GLoVe. And then we could actually have a much stronger prior. If the paper is accepted, I think such an experiment would be very informative.

---

> ### Author Response · Authors · 2020-11-19
> **Response to reviewer 3 (2/2)**
>
> ### Probe sensitivity for sentiment analysis
> * We believe there is a misunderstanding of what baseline means in this setting:
>     * The baselines for probe sensitivity are NOT models, they are the embeddings being probed by these models.
>     * A reasonable baseline here should be a set of embedding without sentiment information. Again we use the LINEAR embedding baseline.
> * And we indeed did not add such baseline in the original version. So we have added new results in Table 7 (Updated Appendix). The accuracy is 48.4 for both Euclidean and Poincaré probes for the baseline LINEAR embedding, meaning that they are doing nearly random guess (near 50 accuracy) and neither of them forms a classifier.
>
> ### Statistical Significance
> Thank you for asking, below is the table of p-values:
>
> |             | Distance  |       |            |       |         |         |
> |-------------|-----------|-------|------------|-------|---------|---------|
> |             | Euclidean |       | Poincaré   |       | p-value |         |
> | Method      | UUAS      | DSpr. | UUAS       | DSpr. | UUAS    | DSpr.   |
> | ELMo0       | 26.8      | 0.44  | 25.8       | 0.44  | 1.3e-7  | 0.41    |
> | Linear      | 48.6      | 0.58  | 45.7       | 0.58  | 1.1e-6  | 0.18    |
> | ELMo1       | 77.0      | 0.83  | 79.8       | 0.87  | 5.4e-3  | 7.7e-4  |
> | BERTbase7   | 79.8      | 0.84  | 83.7       | 0.88  | 9.9e-10 | 4.6e-8  |
> | BERTlarge15 | 82.1      | 0.86  | 85.1       | 0.89  | 6.5e-10 | 4.4e-10 |
> | BERTlarge16 | 81.9      | 0.87  | 85.9       | 0.90  | 1.6e-9  | 3.2e-12 |
>
> |             | Depth     |       |            |       |         |        |
> |-------------|-----------|-------|------------|-------|---------|--------|
> |             | Euclidean |       | Poincaré   |       | p-value |        |
> | Method      | Root %    | NSpr. | Root %     | NSpr. | Root %  | NSpr.  |
> | ELMo0       | 54.2      | 0.55  | 53.5       | 0.49  | 0.010   | 9.2e-5 |
> | Linear      | 2.9       | 0.26  | 4.5        | 0.26  | 0.37    | 0.52   |
> | ELMo1       | 86.5      | 0.86  | 88.4       | 0.87  | 1.5e-6  | 3.9e-3 |
> | BERTbase7   | 88.2      | 0.87  | 91.3       | 0.88  | 7.2e-3  | 0.013  |
> | BERTlarge15 | 89.0      | 0.88  | 91.1       | 0.88  | 7.4e-3  | 0.043  |
> | BERTlarge16 | 89.6      | 0.88  | 91.7       | 0.89  | 6.3e-5  | 5.0e-3 |
>
> We see most of the results hold for the UUAS metrics (the most prominent metrics as discussed previously).
>
> ### Explaining scores in Table 1 and referring labels for subfigures in figure 6
> Thank you for pointing this out, we have updated the corresponding section 4.1 and 5.2 in the updated paper.

---

> ### Author Response · Authors · 2020-11-19
> **Response to reviewer 3 (1/2)**
>
> We thank the reviewer for the detailed opinions, here are our responses:
>
> ### Clarification on probe sensitivity
>
> * The reason that we want a sensitive probe is that, it gives an accurate estimate of the (intractable) real syntactical capability of contextualized embeddings, which is crucial for the community to assess what these embeddings have learned and what they can do.
> * We would like to take this chance and discuss how to define a sensitive probe in the most rigorous notion by doing a Bayesian style thought experiment under an ideal setting:
>     * Suppose we can manually construct two sets of baseline embeddings E0 and E1 and _strictly control_ that the UUAS for E0 to be exactly 25 (= little syntax information) and the UUAS for E1 to exactly 80 (= rich syntax information).
>     * Then we construct a probe to and test the UUAS from that probe. We would like the outputs to be close to 25 and 80 respectively. In this ideal setting, we can exactly measure the sensitivity by the error to the true score.
>     * Such ideal setting is based on the assumption that we know the true UUAS from the very beginning. However it is _currently impossible_ to construct such two sets of embeddings and strictly control the UUAS. Also such golden UUAS prior information is _intractable_ for the ELMo0 and BERTbase7 embeddings, i.e.,we can never know exactly what the numbers are, otherwise we can use the errors to measure sensitivity.
>     * So we _approximate the unknown prior with belief supported by strong evidence_: we hold strong belief that ELMo0 should contain as little syntax as possible, and BERTbase7 should have strong syntax information. Such belief comes from abundant theoretical and empirical evidences about BERT's syntactical capability (and ELMo0's incapability). Note that such belief is also strongly yet implicitly hold by Hewitt and Manning (2019), and serves as a foundation of their work.
>     * In our setting, since there are also abundant theoretical and empirical evidences showing the Euclidean probe's incapability of modeling trees, i.e., there should be more tree information that cannot be correctly revealed in the Euclidean space. To correctly recover the tree information, we need a probe with better inductive bias for trees, this is why we propose the Poincaré probe.
>     * The Poincaré probe gives lower UUAS estimate on ELMo0 (meaning its not a better parser) and higher UUAS on BERTbase 7 (Table 1), where the major gain is from deeper trees (Figure 2 C). We further highlight additional statistics  (Figure 3 in the updated paper) showing the edge length distribution of the Poincaré probe is closer to the ground truth distribution than its Euclidean counterpart.
> * All these inductive bias, theoretical properties, and empirical results serves as strong evidence that Poincaré probe reveals more syntactical information for deeper trees encoded in BERT embeddings (i.e., being more sensitive to the existence of deeper syntax). So we hold strong belief that BERT is underestimated by Euclidean probes.
> * Yet we choose to be careful and not to conclude that BERT is geometrically hyperbolic. Current evidence in this paper may make this hypothesis stronger, but not strong enough to make us conclusive (as is also discussed in section 4.3).
> * We indeed do not aim to be conclusive about what geometry of BERT should be. Acknowledging the fact that the community is still far from thoroughly understanding contextualized embeddings (and more general blackbox neural networks), our goal is to use as scientifically rigorous language as we can to describe our exploration and findings of the Poincaré probe, and inspire the community to find more, e.g., to study contextualized embeddings from a more geometric perspective, or to explore more about hyperbolic geometry like using them to construct better parsers.
>
> ### Interpretation of scores on Table 1
>
> * We note that the depth probing task itself is not as challenging as distance because the later is to estimate pairwise relation while the former is unary, especially for the NSpr metrics (same argument holds for the original Hewitt and Manning 2019). The UUAS should be the most representative metrics as it measures the reconstruction of the whole tree.
> * The highlight of Table 1 is that Poincaré probe gives lower UUAS for ELMo0 than the Euclidean (where we hold strong belief that these embeddings should contain little syntax), meaning that it does not form a better parser (as opposed to the GRU in Appendix A, which effectively make the model a parser rather than a probe); it also gives higher UUAS for BERTbase7 where the major gain is from longer sentences and deeper trees (Figure 2C and Figure 3), which reveals deeper syntactical capability of BERT.

---

### Official Review · AnonReviewer4 · 2020-10-27
**A good argument for taking the curvature of the space into account for probes, but the comparison might not be entirely fair and the experimental setting description needs to be more complete.**

**Rating:** 5
**Confidence:** 4

**Review:**

## Summary

This work examines some of the syntactic and semantic information present in contextual word embeddings by training probes for dependency parsing and sentiment classification. The probes take the form of a low-dimensional projection of the word embeddings, and obtain the dependency parse and sentiment by considering the distance between the pairs of embeddings or between the embeddings and a class embedding respectively.

This is closely related to the approach of (Hewitt and Manning, 2019), whose experimental setting this work reproduces, but the innovation here consists in using hyperbolic rather than Euclidean distances: the embedding space is identified with the tangent space at the origin and projected to the Poincare ball using its exponential map, allowing the probe to use the hyperbolic distance.

The hyperbolic version of the probes consistently out-perform the Euclidean one, which is encouraging, and the authors provide some useful visualizations of the learned projections. However, the experiments fail to account for one possibly relevant difference between their Hyperbolic and Euclidean setting. The paper is also at times difficult to follow on its own, as it relies a bit too much on cited work.

## Clarity

Even if the experimental setting is the same as (Hewitt and Manning, 2019), a quick summary would be welcome here: currently, the name of the parsing dataset is not even mentioned, nor are the meanings of the Spearman correlation metrics (the abbreviations are not self-explanatory).

There are also some open questions: for parsing, do you train a single model with both objectives? From reading the code, it looks like you alternate doing an epoch of each. Is that right? How does that compare to learning both individually?

## Correctness

My main issue with the comparison lies in the second projection which the hyperbolic (but not the Euclidean) method uses, as defined in Equation (3)

First, doing the projection in the hyperbolic space is redundant. Remember that the Mobius matrix-vector simply applies the logarithmic map at the origin followed by a linear transformation in the tangent space followed by the exponential map at the origin. Hence, Equations (2) and (3) could simply be summarized as:
$$q_i = \exp_0 (Q P h_i) $$

(Same remark for the GRU in the appendix.)

While this would be equivalent to having a single projection matrix if we could do global optimization, given the geometry of the loss function the hyperbolic setting with its additional parameter might be easier to optimize than the Euclidean one. In particular, this probably accounts for the difference in sentiment classification with fixed label embeddings.

Thankfully, the difference in the same task with learned embedding seems to indicate that there is more to the result, and I would consider increasing my score if all the results hold with either an additional linear transformation in the Euclidean setting or a single projection in the hyperbolic.

---

> ### Author Response · Authors · 2020-11-19
> **Response to reviewer 4**
>
> We thank the reviewer for the detailed opinions, here are our responses:
>
> ### Adding an additional linear transformation to the Euclidean probe
>
> * We agree that the second projection makes the configuration of the Poincaré probe different than the Euclidean one. And yes from the equations, this projection is redundant. We do it primarily for optimization purpose (as you have noticed) because the numerics at the boundary of the Poincaré ball might be unstable and inaccurate during gradient descent (a common issue for hyperbolic models, Nickel and Kiela 2017, Ganea et. al. 2018; also the same reason for GRU).
> * We have added a new experiment about additional linear transformation in the Euclidean probe. The new results are (detailed in updated Appendix Table 9):
>     * Euclidean two layers (new results): BERTbase7: UUAS 79.9, DSpr 0.84 | ELMo0: UUAS 26.7, DSpr 0.44
>     * Euclidean one layer (Table 1 in the paper): BERTbase7: UUAS 79.8, DSpr 0.85 | ELMo0: UUAS 26.8, DSpr 0.44
>     * Hyperbolic (Table 1 in the paper): BERTbase7: UUAS 83.7, DSpr 0.88 | ELMo0: UUAS 25.8, DSpr 0.44
> * We see that adding new linear layer to the Euclidean probe barely changes its performance and rank of the probes still holds.
> * Additionally, we have changed the Hyperbolic probe to one linear transformer, the results on BERTbase7 do decrease to: 80.2 UUAS, 0.87 DSpr.
>     * Though it is still higher than the two Euclidean probes, we would not conclude the margin is large enough
>     * However we further note this is primarily for optimization reasons as mentioned above, i.e., in this setting, the Poincaré probe are not optimized to its full potential.
> * Additionally, we have also added non-linearity to the Euclidean probes (per Reviewer 1's request, detailed in the responses to them).
>     * There are two consequences after doing it: (a) this would effectively change the geometric structure of the Euclidean probe, making it not strictly Euclidean (b) the probing scores under this setting would also increase for the ELMo0 embeddings (which we do not expect).
>     * Our conclusion is that adding non-linearity to Euclidean probes would make it more like a parser, rather than a more sensitive probe.
> * Despite these additional results, we would like to highlight that our primary results hold when comparing a Euclidean probe with two linear transformations with the Poincaré probe.
>
> ### Clarity
>
> * Thank you for pointing this out. We use the PTB dataset with Stanford Dependency formalism. Spearman correlation means how the recovered distance/ depth correlate to the golden label. We have updated more details in the section 4.1 of the new version.
> * The numbers reported in Table 1 and Figure 2 are from two models on two objectives separately (code config/example*.yaml, task = distance or depth or both). We do train a single model with both two objectives added together (results in Appendix Table 4, slightly worse than separate training). The primary reason for joint training is for better visualization: we want to see trees with roots close to the origin and see how it expands to the edge of the space.  This kind of illustration would help us better understand the geometric structure. Figure 1 and 3 in the main paper and all tree visualizations in the appendix are from joint training.
>
> ### References
>
> * Maximillian Nickel and Douwe, Kiela. *Poincaré Embeddings for Learning Hierarchical Representations*. NeurIPS 2017
>
> * Octavian Ganea and Gary Becigneul and Thomas Hofmann. *Hyperbolic Neural Networks*. NeurIPS 2018

---

### Official Review · AnonReviewer2 · 2020-10-28
**Simple, interesting, could stand more rigor**

**Rating:** 7
**Confidence:** 3

**Review:**

#### Summary:

In the same vein as Hewitt & Manning 2019, the authors present an extremely lightly parametrized “probe” model to determine the presence of syntactic structure in the embedding space of BERT models. While Hewitt & Manning examine the Euclidean distance between linearly transformed token embeddings and its correlation with parse tree distance and depth, this work examines a different distance function based on distances in hyperbolic space. They find that this distance measure, for an equivalent or lesser number of parameters, better reproduces the syntactic properties. This suggests that the BERT model may operate simply, but on a non-Euclidean manifold, in order to work with syntactic information.


#### Reasons for score:


The paper presents a simple method to examine syntactic structure in contextual embedding models, as a nice extension of the seminal work on Euclidean probing, and should be directly of interest to the ICLR audience. The paper seems to be well-executed experimentally, with a variety of analyses.

However, I would like to see the authors give more geometric rigor to their model and argument, and clarify the hypothesis which they are testing. There is a lot of appeal to intuition about hyperbolic space and its ability to encode trees, but it is not clear what that has to do with their proposed probe in a concrete way.

The paper would be greatly improved by a simple 1 to 3 dimensional example of their proposed probe and how it could discover an embedded submanifold with hyperbolic structure.

#### Positives:

- This is a mathematically (and superficially conceptually), simple generalization of the work of Hewitt & Manning, which offers improvements to and new avenues for exploration of these probing ideas.
- The work convincingly shows that their probe offers better performance, especially in UUAS, compared to Euclidean probes, which is quite interesting.
- Generally, I believe that there is a good idea here, but it would be much improved by additional mathematical rigor.

#### Comments/Concerns:

- It is hard to understand this paper without having read Hewitt & Manning, it could do with being a bit more self-contained. For example, the metrics like “DSpr”, etc, should be explained.

- It is not clear what hypothesis is being tested here. What does it mean for the embedding space to have a hyperbolic structure in this sense? The Nash embedding theorem ensures the existence of an isometric embedding of d-1 dimensional hyperbolic space in d dimensional Euclidean space, but that embedding can hardly be accomplished in general by the simple linear transform + exponential map + moebius multiplication described here.

- In this vein, there should be much more rigor as to describing what exactly the Poincaré probe is doing. The paper talks about mapping to “the tangent space”. What tangent space is this? Is it the tangent space of a Poincaré ball at the origin? Why is this a reasonable thing to do if the goal is to discover hyperbolic structure? When should this be expected to work?

- An example in low dimensions with visuals of a hyperbolic submanifold, and a Poincaré probe interacting with it, should feature prominently.

- Along these same lines, it would be very helpful for the paper to give an example of how the neural network might access this information on a hyperbolic submanifold, though this may be beyond the scope of the work.

- In the introduction, the authors note the ability of hyperbolic space to encode exponentially larger volumes vs Euclidean space, but of course the Poincaré ball they are examining is embedded in Euclidean space.  What does this mean concretely in the context of the BERT transformer network?

#### Minor comments:

- middle of page 7, “RiemannianAdam” is missing the space

---

> ### Author Response · Authors · 2020-11-21
> **Response to reviewer 2**
>
> We thank the reviewer for the detailed comments, here are our responses:
>
> ### Updated description of the hypothesis and the Poincaré probe
> * First, we note that there could be a misunderstanding as the reviewer is discussing the isometric embedding of the d-1 dimensional hyperbolic space in the d dimensional Euclidean space. We want to clarify that:
>     * We are not trying to find an isometry between the hyperbolic and the Euclidean (original BERT space), we are trying to find an (approximately) isometry between the hyperbolic space and the raw tree space where the elements in this space are nodes and the distance is number of edges from one node to another.
>     * So a formal statement of the hypothesis here is: (a) whether there exists a Poincaré ball (transformed from the original BERT embedding space with simple parameterization) that encodes dependency trees (in the same vein with Hewitt and Manning 2019), and further (b) whether such Poincaré probes are more sensitive to the existences of deeper syntax (as the property induced from its own inductive bias, also discussed in our response to Reviewer 3).
> * More detailed explanation of probe design. We note that our probe is now: $q = Q exp_0(Ph)$, below we explain each term in detail:
>     * $exp_0$ and $Ph$: the use of the exponential map at the origin follows previous work in Genea et. al. (2018), Mathieu et. al. (2019). The primary reasons are (a) simpler mathematical formulation; (b) simpler optimization. Consequently, $Ph$ is a vector in the tangent space of the origin of a Poincaré ball and $P$ is a Euclidean-to-Euclidean mapping from the BERT original space (assumed to be Euclidean) to the tangent space. Though our choice follows previous works, we do agree that a more in-depth study, either from a geometric perspective or from a linguistic perspective, is worth investigating.
>     * The second mapping $Q$ is also primarily for optimization reason: we empirically observe that the numerics would be unstable without this mapping (note that the numerical stability is still an open issue for hyperbolic deep learning, Nickel and Kiela 2017, Becigneul and Ganea 2019) and it essentially maps the embeddings from one Poincaré ball to another. Also note the reviewer 1 and 4 pointed out this mapping could possibly make the comparison to the Euclidean counterparts not that fair, so we updated the Euclidean probe to make a more fair comparison (see details in responses to them) and the results are basically consistent with the previous results.
>
> ### Low dimensional example
> * Thank you for your suggestion. We have added examples about changes of shapes and localizations of probed trees with regard to the curvature of the Poincaré ball in the updated paper (Figure 18-20, Table 10).
> * Basically, as the curvature of the Poincaré ball changes to be closer to -1, the trees would be more close and stretched to the edge of the ball with increasing syntax recovering performance.
>
> ### Interpretation of the hyperbolic subspace under the context of BERT transformer network and further comments
>
> * First, we assume that the BERT original space is (or embedded in) a Euclidean space. The contextualized word embeddings within this space lay in certain special manifolds whose structures and meanings we are currently not aware. We note that these manifolds should encode more information than syntax.
> * Then our Poincaré distance probe maps the embeddings manifolds (which we know little about) to a Poincaré ball, under which the tree distances are approximately equal to the squared hyperbolic distance (similar arguments apply to the depth probe).
> *  This Poincaré probe essentially disentangles syntactical information out of the mixed information encoded in the original embedding, so we also call the Poincaré ball a syntactical subspace.
> * Although we know more about the properties of the Poincaré ball, we do agree that the properties of the embedding manifold in the original BERT embeddings space, how the information flow and transform along these spaces, and how BERT access these information in a hyperbolic (or non-hyperbolic) way should be more deeply investigated. We leave them to future work.
>
> ### References
> * Octavian-Eugen Ganea, Gary Bécigneul and Thomas Hofmann. *Hyperbolic Neural Networks*. NeurIPS 2018
> * Emile Mathieu, Charline Le Lan, Chris J. Maddison, Ryota Tomioka, Yee Whye Teh. *Continuous Hierarchical Representations with Poincaré́ Variational Auto-Encoders*. NeurIPS 2019
> * Gary Bécigneul and Octavian-Eugen Ganea. *Riemannian Adaptive Optimisation Methods*. ICLR 2019
> * Maximilian Nickel and Douwe Kiela. *Poincaré Embeddings for Learning Hierarchical Representations*. NeurIPS 2017

---

### Official Review · AnonReviewer1 · 2020-10-28

**Rating:** 6
**Confidence:** 3

**Review:**

This paper proposes probing BERT representations by projecting them into a Poincare subspace. The proposed approach is used to probe ELMO and BERT for both syntax and sentiment in comparison with the conventional Euclidean probes.

I am ambivalent about this paper. On the positive side, I think that it is a quite solid work, with extensive experimentation, additional supporting results in the appendix, and an accompanying code that can be used to reproduce results and obtain additional visualizations. The paper is also well written and the authors are rigorous when discussing their results rather than trying to oversell.

On the negative side, I have some reservations about the relevance of this study. What do we learn from it? It is true that the Poincare probes obtain generally higher scores than the Euclidean probes, but it doesn't look like they lead to any new insight about how BERT works. If the message here is that Poincare probes are more appropriate than their Euclidean counterparts, I would have liked to see instances were Euclidean probes lead to erroneous or at least different conclusions when compared to Poincare probes. In the absence of that, we can expect that practitioners will stick with Euclidean probes given that they are simply easier to use.

Moreover, I am not sure if the comparison with Euclidean probes is entirely fair. If my understanding is correct, the Poincare probes learn two linear transformations (P and Q), whereas Euclidean probes learn a single one. Unless I am missing something, it could be that the Poincare probes obtain higher scores simply because the transformation they are learning is more expressive, and not because of the underlying geometric space. In order to test this hypothesis, I think that the authors should try learning two linear transformations for Euclidean probes, with a non-linearity like ReLU in between.

Finally, I feel that some of the analyses did not follow a systematic methodology and some of the interpretations seem subjective and possibly questionable. For instance, I don't see any clear difference between the Poincare and Euclidean probes when it comes to the sentence length (Figure 2C), except for very short sentences. For sentence length > 12 the curves look very similar to me, except that the absolute values for Poincare are higher. Similarly, the visualizations, although interesting, provide a rather anecdotal evidence, in particular since the examples seem to be cherry-picked.

---

> ### Author Response · Authors · 2020-11-20
> **Response to reviewer 1 (2/2)**
>
> ### Changing Euclidean probe to two linear transformations with a non-linearity
> * We agree that extending the Euclidean probe to two layers would make it more fair comparing to the Hyperbolic Probe. The additional results are in Table 9 in the updated Appendix.
> * Specifically, if we further add Sigmoid non-linearity to the two-layer Euclidean probe, the UUAS score on BERTbase7 would be 84.5 (larger than Poincaré = 83.7). However its UUAS score on the baseline ELMo0 also increases to 29.8 (larger than the previous one-layer Euclidean = 26.8, also larger than Poincaré = 25.8).
> * Our interpretations on the results of two-layer Euclidean probe with non-linearity are:
>     * Although it gives higher UUAS than Poincaré for BERTbase7, it also increases the UUAS on the baseline ELMo0 (while a sensitive probe should give lower UUAS for ELMo0), meaning its parsing capability is indeed increased with the sigmoid function.
>     * The sigmoid function essentially changes the underlying geometric structure since it concentrates all values into [0, 1], so it may transform the original space into another space with (currently) unclear geometry. While we do think there are many fine-grained nuance under each space that definitely worth more digging, we believe these results do not necessarily indicate that our Poincaré probe is less sensitive than the Euclidean counterpart.
> * Additionally, we also include the results on (a) two-layer Euclidean probe without non-linearity (per reviewer 4's request) / with tanh/ ReLU (b) probing results of all these probes applied to a new RANDOM baseline embeddings (where the probed embedding is just a randomly sampled matrix). We have:
>   * Our Poincaré probe shows better sensitivity than the two-layer Euclidean probe without non-linearity  (see detailed discussions in the response to reviewer 4).
>   * The results for two-layer Euclidean probes with tanh and ReLU show that they also behave more like a parser.
>   * Arguments similar to probing results on the ELMo0 baseline apply to results on the RANDOM baseline.
> * In conclusion, we would say that a two-layer Euclidean probe is more like a parser, rather than a more sensitive probe.

---

> ### Author Response · Authors · 2020-11-20
> **Response to reviewer 1 (1/2)**
>
> We thank the reviewer for the detailed opinions, here are our responses:
>
> ### Cases where Euclidean and Poincaré are different and what do we learn from them
> * We would draw the reviewer's attention on the following aspects showing the differences of the two probes:
>     * Top sentiment words from the two probes (Table 3): though we have a lot of attention on the syntax task, firstly we note that in the sentiment task, these words are not cherry-picked, they are strictly ranked by the probes. This would serve as the first case showing the differences between the two probes where the Poincaré probe gives more words align with human intuition.
>     * (New results): Randomly sampled sentences (Figure 11 in the updated Appendix): the current qualitative comparison in the main paper Figure 4 (previously Figure 3 in old version) is indeed cherry-picked for illustration purpose. So we add 12 randomly sampled sentences with different length in the Appendix as a more detailed qualitative analysis. We roughly observe that the major differences are about edges spanning long sentence chunks, which motivate our next new experiment.
>     * (New results): following the previous observation that the differences of the two probes are on the long edges, we compare the edge length distributions of golden trees and trees produced from the two probes. The results are updated in Figure 3 in the paper main body as a bar chart. Here we give the detailed statistics of the number of edges recovered by these probes (order by edge length in 1-5, 6-10, 11-15, 16-20, >20):
>         * Ground truth: 30420 (90.30%), 2124 (6.30%), 695 (2.06%), 255 (0.75%), 192 (0.56%)
>         * Poincaré: 30752 (91.29), 1931 (5.73%), 642 (1.90%), 217 (0.64%), 144 (0.42%)
>         * Euclidean: 30778 (91.36%), 2001 (5.94%), 612 (1.81%), 183 (0.54%), 112 (0.33%)
>         * We see that both two probes underestimate the number of long edges, which verifies that deeper trees are indeed more challenging to discover. yet the distribution of the Poincaré probe is closer to the ground truth than the Euclidean probe while the later the more biased towards short edges. These results would serve as another evidence that the Poincaré probe better recovers deeper syntax.
>     * UUAS w.r.t. sentence length in Figure 2C: to interpret the results, we would emphasize that it is the absolute values that reveal where the performance gain comes from: the Poincaré probe consistently better recovers the syntactical information (higher UUAS) on long edges than the Euclidean counterpart. These results also align with our new experiments showing that the Poincaré probe gives a closer edge length distribution than the Euclidean counterpart.
> * Combining these four differences, we would advocate using our Poincaré probes to the practitioners for its sensitivity to the existence of deeper syntax:
>     * A major reason that we want a more sensitive probe is that it gives a more accurate estimate of the underlying syntactical capability of contextualized embeddings, which is crucial for the community to assess what these embeddings have learned and what they can do. It would also help the community to study the geometric structure of contextualized embeddings from a hyperbolic machine learning perspective.
>     * Furthermore, the sensitivity for longer edges would be particularly important from a linguistic perspective because: (a) longer edges reveal more complicated dependency structures in sentences (b) longer edges are also the most often places where existing models make wrong predictions, as is commonly observed in parsing and structured prediction literature, which further leads to decades of targeted works.
>     * We also believe that in our implementation, the Poincaré probe is just as easy to use as the Euclidean counterpart (partially due to its own simplicity). In the attached Jupiter notebook we further provide abundant visualization. We would encourage the audience to try it out.

---

### Decision · Program_Chairs · 2021-01-07
**Final Decision**

**Decision:**

Accept (Poster)

**Comment:**

The paper introduces a new method to probe contextualized word embeddings for syntax and sentiment properties using hyperbolic geometry. The paper is written well and relevant to the ICLR community. Reviewers highlight that the proposed Poincaré probe offers solid results, extensive experiments that support the benefits of the approach, and proposes a new approach to analyze the geometry of BERT models. The revised version clarified various concerns of the initial reviews and improved the manuscript (comparison to Euclidean probes, low dimensional examples, new results on edge length distributions etc.). Overall, the paper makes valuable contributions to probing contextualized word embeddings and the majority of reviewers and the AC support acceptance for its contributions. Please revise your paper to take feedback from reviewers after rebuttal into account (especially to further improve clarity and discussion of the method).